# A bead-based GPCR phosphorylation immunoassay for high-throughput ligand profiling and GRK inhibitor screening

Johanna Kaufmann[1,2,7], Nina Kathleen Blum[1,2,7], Falko Nagel[2], Anna Schuler[1], Julia Drube [3], Carsten Degenhart[4], Julian Engel[4], Jan Eicke Eickhoff[4], Pooja Dasgupta[1], Sebastian Fritzwanker[1], Maria Guastadisegni[5], Clemens Schulte[6], Elke Miess-Tanneberg[1], Hans Michael Maric [6], Mariana Spetea [5], Andrea Kliewer [1], Matthias Baumann[4], Bert Klebl[4], Rainer K. Reinscheid[1], Carsten Hoffmann [3] & Stefan Schulz [1,2✉]

Analysis of agonist-driven phosphorylation of G protein-coupled receptors (GPCRs) can provide valuable insights into the receptor activation state and ligand pharmacology. However, to date, assessment of GPCR phosphorylation using high-throughput applications has been challenging. We have developed and validated a bead-based immunoassay for the quantitative assessment of agonist-induced GPCR phosphorylation that can be performed entirely in multiwell cell culture plates. The assay involves immunoprecipitation of affinity-tagged receptors using magnetic beads followed by protein detection using phosphorylation state-specific and phosphorylation state-independent anti-GPCR antibodies. As proof of concept, five prototypical GPCRs (MOP, C5a1, D1, SST2, CB2) were treated with different agonizts and antagonists, and concentration-response curves were generated. We then extended our approach to establish selective cellular GPCR kinase (GRK) inhibitor assays, which led to the rapid identification of a selective GRK5/6 inhibitor (LDC8988) and a highly potent pan-GRK inhibitor (LDC9728). In conclusion, this versatile GPCR phosphorylation assay can be used extensively for ligand profiling and inhibitor screening.

[1] Institut für Pharmakologie und Toxikologie, Friedrich-Schiller-Universität Jena, Universitätsklinikum Jena, Jena, Germany. [2] 7TM Antibodies GmbH, Jena, Germany. [3] Institut für Molekulare Zellbiologie, CMB—Center for Molecular Biomedicine, Friedrich-Schiller-Universität Jena, Universitätsklinikum Jena, Jena, Germany. [4] Lead Discovery Center GmbH, Dortmund, Germany. [5] Department of Pharmaceutical Chemistry, Institute of Pharmacy and Center for Molecular Biosciences Innsbruck (CMBI), University of Innsbruck, Innsbruck, Austria. [6] Rudolf Virchow Center, Center for Integrative and Translational Bioimaging, University of Wuerzburg, Wuerzburg, Germany. [7] These authors contributed equally: Johanna Kaufmann, Nina Kathleen Blum. ✉email: stefan.schulz@med.uni-jena.de

Gprotein-coupled receptors (GPCRs) are vital signal transducers that mediate the effects of a variety of chemical and physical stimuli, and represent major drug targets for many human diseases. GPCRs have a uniform structure with seven-transmembrane domains and are therefore also referred to as seven-transmembrane receptors (7TMRs). Activation by their endogenous ligands or exogenous agonizts leads to GPCR conformational changes that are recognized by a family of kinases termed G protein-coupled receptor kinases (GRKs)[1]. The unique ability of GRKs to recognize activated receptors results in agonist-dependent phosphorylation at intracellular serine and threonine residues[2,3]. In addition, second messenger-regulated kinases can also participate in GPCR phosphorylation, including protein kinases A and C and other serine/threonine kinases[4-6]. Phosphorylation of GPCRs is a biologically and pharmacologically important process that primarily initiates desensitization and internalization of a receptor[7,8]. Phosphorylation also increases the interaction of GPCRs with intracellular adapter proteins such as β-arrestins, which can trigger a second wave of signaling[9-12].

Initial GPCR studies were based on radioactive whole-cell phosphorylation assays, which require high levels of radioactivity and cannot be used to identify individual phosphorylated serine and threonine residues. Later, studies were directed to phosphoproteomics analysis, through which limited quantitative information has been obtained[13]. Currently, the prevailing approach is based on antibodies that specifically recognize the phosphorylated state of a GPCR[14-18]. When available, phosphosite-specific anti-GPCR antibodies are excellent tools to resolve the temporal and spatial dynamics of receptor phosphorylation, identify relevant kinases and phosphatases, detect receptor activation and profile the pharmacological properties of new ligands[19-22]. However, phosphosite-specific antibodies are predominantly used in immunoblotting approaches that are technically challenging, limited to small sample sizes and difficult to quantify.

In pharmaceutical and academic research, high-throughput assays are often required to facilitate the screening of large chemical libraries. Although receptor binding, G-protein signaling and arrestin recruitment can be assessed with available methodologies, adequate technologies for the determination of GPCR phosphorylation are currently not available. We were therefore motivated to take advantage of phosphosite-specific anti-GPCR antibodies in the development of a quantitative phosphorylation immunoassay that can be performed entirely in multiwell cell culture plates and is conducive to medium to high-throughput applications (Fig. 1). Because this assay can be adapted to virtually all seven-transmembrane receptors, it is referred to as '7TM phosphorylation assay'.

## Results

### Development of a 7TM phosphorylation assay

This study was performed to develop a quantitative GPCR phosphorylation immunoassay. To this end, HEK293 cells stably expressing hemagglutinin (HA)-tagged mouse μ-opioid receptors (MOPs) were seeded into poly-L-lysine-coated F-bottom 96-well plates and grown to >95% confluency. The cells were cultured in the presence or absence of the agonist [D-Ala$^2$,N-MePhe$^4$,Gly-ol]-enkephalin (DAMGO) and then lysed in detergent buffer containing protease and protein phosphatase (PPase) inhibitors. After plate centrifugation, lysates were transferred to U-bottom 96-well plates. MOPs were then immunoprecipitated using mouse anti-HA antibody-coated magnetic beads. After washing the beads under magnetic force, either phosphosite-specific rabbit antibodies that specifically recognized the S375-phosphorylated

form of MOP (pS375-MOP) or antibodies that detected MOP in a phosphorylation-independent manner (np-MOP) were added to the cells (Fig. 2a). Primary antibody binding was then detected using commercially available enzyme-labeled secondary antibodies followed by addition of the respective enzyme substrate solution. The color reaction in DAMGO-treated wells was stopped when the optical density (OD) read at 405 nm reached 1.2, typically within 2 to 8 min. Under identical conditions, the OD of untreated wells was approximately 0.2, suggesting that DAMGO-induced S375 phosphorylation of MOP was specifically detected within the optimal dynamic range for 2′-azino-bis-(3-ethyl-benzthiazoline-6-sulfonic acid (ABTS)-based enzyme immunoassay substrates (Fig. 2b). Omission of the magnetic beads or the primary or secondary antibodies or the use of untransfected cells resulted in OD readings of 0.2 or less. When DAMGO was added at different concentrations ranging from 10 nM to 10 μM in wells assayed using the anti-pS375-MOP antibody, we observed increasing OD readings with data points following the typical shape of a sigmoidal concentration–response curve (Fig. 2c). In contrast, all wells assayed using the anti-np-MOP antibody yielded an OD reading of ~0.8, independent of agonist exposure, indicating that all the MOPs had been identified (Fig. 2c). Next, we evaluated the effect of PPase inhibitors with both a pS375-MOP phosphorylation immunoassay and western blot assay with cells cultured in a multiwell plate. For the western blot analysis, cells were cultured and treated in 96-well plates as described and lysed in detergent buffer with or without PPase inhibitors. MOPs were immunoprecipitated with anti-HA magnetic beads. Immunoprecipitates were washed under magnetic force, and receptors were eluted from the beads into SDS-sample buffer by incubating the plates for 25 min at 43 °C. When these samples were immunoblotted with the anti-pS375-MOP antibody, a concentration-dependent increase in S375 phosphorylation was observed in samples cultured with PPase inhibitors but not in samples lysed without PPase inhibitors (Fig. 2d, top panel). When the blot was stripped and reprobed with the np-MOP antibody, similar levels of MOPs were detected in all the samples irrespective of the presence of PPase inhibitors or agonist exposure (Fig. 2d, bottom panel). For the in-well assay, cells were plated and grown, treated and lysed as in the western blot assay, except that the samples were directly immunoassayed in the plate. Similar to the results obtained by immunoblotting, the results of the assay performed with the anti-pS375-MOP antibody revealed a concentration-dependent increase in signal intensity only in the wells with PPase inhibitor-treated samples (Fig. 2e). In contrast, in wells assayed with the anti-np-MOP antibody, similar signals were observed independent of inhibitor or agonist treatment (Fig. 2f). Notably, a high degree of correlation between the concentration–response curves obtained through the immunoblot analysis and immunoassay was found (Supplementary Fig. 1). These results show that the addition of PPase inhibitors during cell lysis was an essential component of the assay. These results also show that anti-pS375-MOP antibody binding depended on agonist-induced receptor phosphorylation, which needed to be protected from PPase digestion during cell lysis. The anti-np-MOP antibody bound to receptors regardless of their phosphorylation status, and hence, using this antibody appeared to be a feasible means of controlling total receptor content.

### Validation of the 7TM phosphorylation assay

To substantiate the specific binding of the anti-pS375-MOP antibody, we performed peptide neutralization to introduce controls in the immunoassay. When the anti-pS375-MOP antibody was incubated with an excessive amount immunizing peptide that contained phosphorylated S375, the concentration-dependent

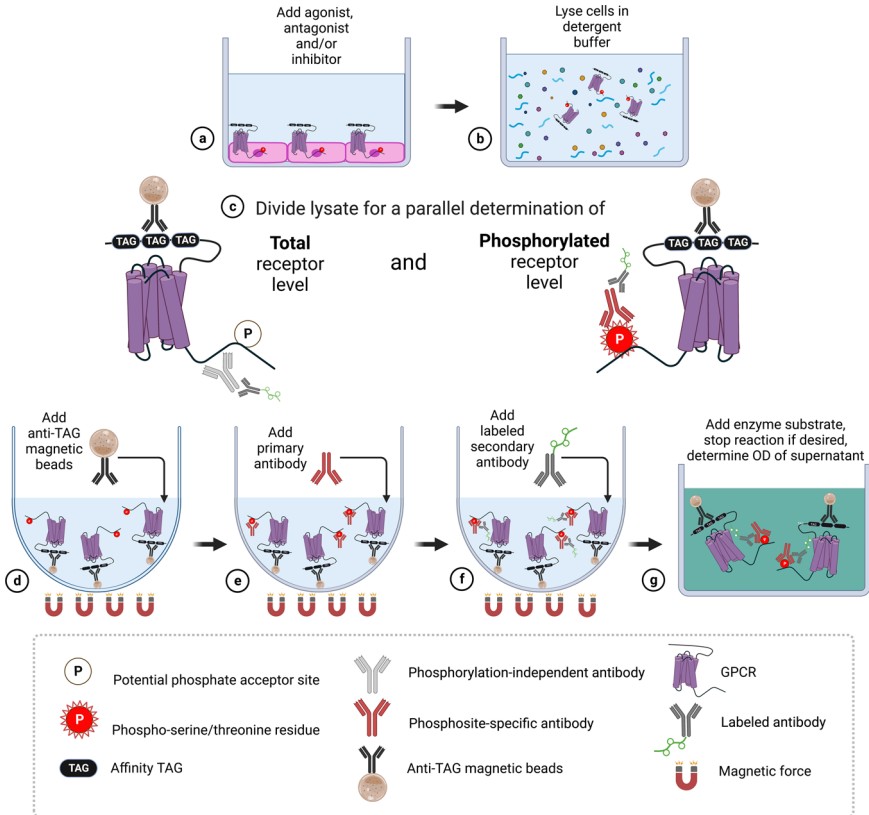

**Fig. 1 Step-by-step flowchart showing the 7TM phosphorylation assay protocol. a** Cells expressing affinity-tagged GPCRs are grown in F-bottom cell culture plates, and upon reaching ≥95% confluency, the cells are exposed to the agonist, antagonist or inhibitor of interest. **b** The cells are lysed in detergent buffer, and lysates are cleared by centrifugation. **c** For parallel detection of phosphorylated and total receptors, the lysate of each sample is divided into corresponding wells in U-bottom assay plates. **d** Anti-tag magnetic beads are added to each well for receptor immunoprecipitation. **e** Primary phosphosite-specific and phosphorylation-independent antibodies are added to the appropriate wells of each split sample. **f** A secondary antibody labeled with an enzyme or other detection entity is then added. **g** An enzyme substrate solution is added for detection, the color reaction is stopped by adding a stop solution, and the optical density (OD) is determined with a microplate reader. (Created with BioRender.com).

increase in signal intensity was completely blocked (Fig. 2g). In contrast, the immunoassay was unaffected by the addition of the corresponding unphosphorylated peptide, indicating unequivocal detection of agonist-dependent S375-MOP phosphorylation under these conditions (Fig. 2g). We next optimized individual assay components such as the quantity of magnetic beads and primary antibody required. The magnetic beads used in this study had a diameter of 1 μm, and 1 μl of the slurry contained 10 μg of these beads. As shown, the addition of 20 μg of magnetic beads per well resulted in high signal intensity with limited background (Fig. 2h). Increasing to 50 μg of magnetic beads did not enhance the signal strength but resulted in a higher background. The signal intensity and the dynamic range of the assay decreased when using 5 μg beads or less (Fig. 2h). We also found that the addition of 4 μg/ml of the anti-pS375-MOP antibody yielded results within the desired signal-to-noise ratio (Fig. 2i). Nevertheless, the optimal quantity of primary antibody needed to be individually determined for each phosphosite-specific antibody. Having established optimal assay conditions, we sought to determine the detection limit. To this end, MOP-transfected cells (MOP-HEK293) and untransfected cells (WT-HEK293) were cultured in separate wells and exposed to saturating concentrations of DAMGO. The MOP-HEK293 cell lysate was stepwise diluted with WT-HEK293 lysate to yield a total volume of 100 μl in each sample, which represented a total of 70,000 cells (Fig. 2j). These treated samples were then evaluated via immunoblot analysis and anti-pS375-MOP phosphorylation immunoassay. In both assays, strong signals were detected in the samples

containing at least 40 to 60 μl of the MOP-HEK293 lysate. Further dilution resulted in a linear decline in signal intensity (Fig. 2j–l). The detection limit of the western blot assays was reached when samples contained less than 20 μl of the MOP-HEK293 lysate (Fig. 2j and Supplementary Fig. 2). In both the pS375-MOP and np-MOP immunoassays, the limit of detection was reached when samples contained less than 5 μl of the MOP-HEK293 lysate (Fig. 2l and Supplementary Fig. 2). Saturation radioligand binding revealed 2279 fmol specific binding sites per mg membrane protein (Supplementary Fig. 3). Based on protein assays 1 mg membrane protein was calculated to represent $4.6 \times 10^6$ cells. Thus, 2279 fmol equal $2279 \times 6 \times 10^8$ receptors per mg membrane protein, representing $4.6 \times 10^6$ cells, which results in 297,260 (~300,000) functional MOP receptors per cell. This allowed us to calculate a detection limit of 80 pg receptor protein for the pS375-MOP immunoassay and 500 pg for the immunoblot assay. In the range of 80 pg to 1200 pg receptor protein, the pS375-MOP immunoassay showed a linear increase in signal intensity (Supplementary Fig. 4). In recovery experiments, the lysates were subjected to a second round of bead-based immunoprecipitation. The results presented in Supplementary Fig. 5 show that the MOPs were not completely removed from the lysates containing more than 40 μl of the original MOP-HEK293 cell lysate. The observation that the pS375-MOP immunoassay yielded linear results using only 50% of the lysate was in each well led us to divide each lysate for the detection of both phosphorylated and total receptor content, thus enabling a true quantitative analysis. Therefore, the step in which the lysate was

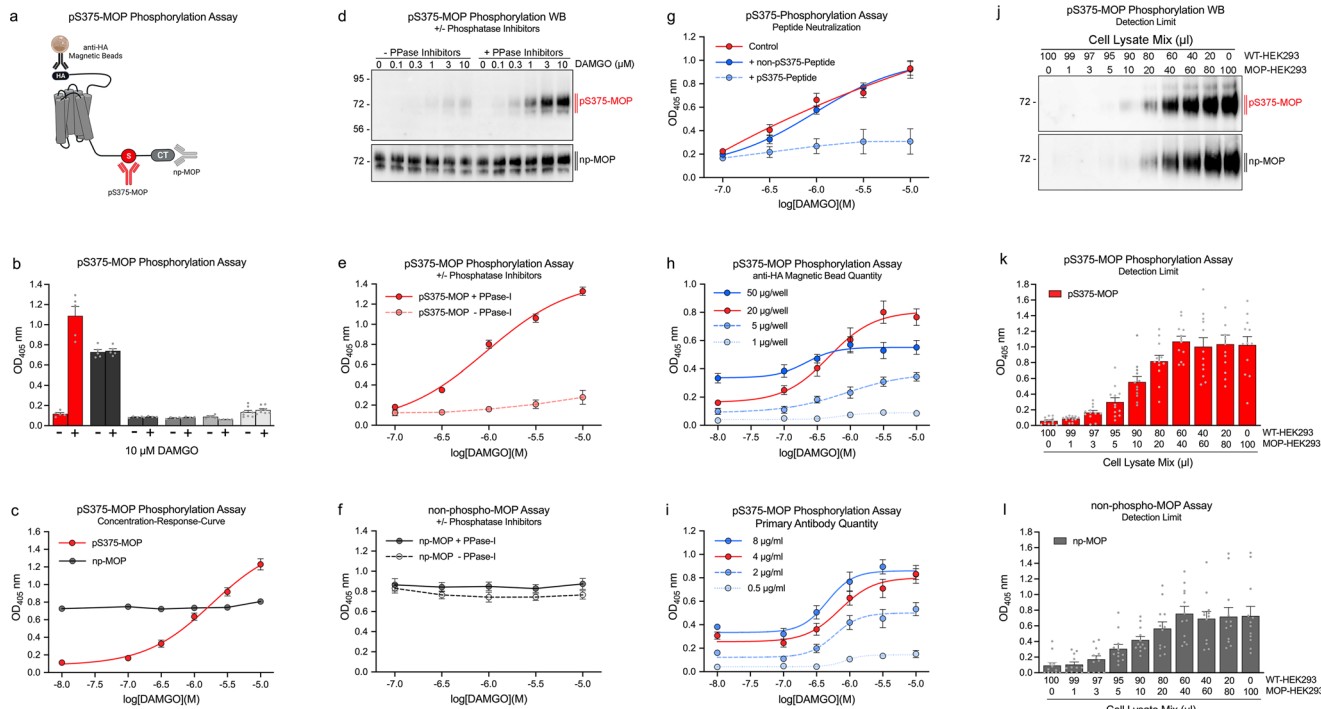

**Fig. 2 Development and validation of the 7TM phosphorylation assay.** In all experiments, the mouse μ-opioid receptor (MOP) was stimulated with DAMGO for 30 min at 37 °C. **a** Schematic representation showing MOP with anti-hemagglutinin (HA) magnetic bead-binding sites, anti-phosphorylated (p) S375-MOP antibody that selectively detects pS375-MOP and an antibody that detects MOP independent of phosphorylation status (np-MOP). **b** Comparison of optical density (OD) values under different assay conditions: stimulated (+) or unstimulated (−), no primary antibody, no secondary antibody, no beads with MOP-transfected or untransfected (WT)-HEK293 cells. An anti-pS375-MOP was used as the primary antibody in all conditions except for the np-MOP samples. **c** The concentration–response OD readings with increasing DAMGO concentrations were determined with anti-pS375-MOP and anti-np-MOP antibodies. **d**–**f** MOP-HEK293 cells were treated with increasing DAMGO concentrations. Cells were lysed in detergent buffer in the presence or absence of protein phosphatase inhibitors (±PPase-I) and either analyzed by western blotting (**d**), pS375-MOP assay (**f**) or np-MOP phosphorylation assay (**e**). **g** For peptide neutralization, the anti-pS375-MOP antibody was preincubated with 1 μg/ml of pS375 peptide (solid blue line) or the corresponding unphosphorylated peptide (dashed blue line) for 1 h before the addition of each antibody corresponding to the beads. Control samples (solid red line) were prepared according to a standard assay protocol. (**h**) Comparison of DAMGO concentration–response curves after the addition of different amounts of anti-HA magnetic beads, ranging from 1–50 μg per well. The optimal amount is depicted as a red solid line. **i** Comparison of DAMGO concentration–response curves after the addition of different amounts of primary antibody, ranging from 0.5 to 8 μg/ml. The optimal amount is depicted as a red solid line. **j**–**l** For determination of detection limits, lysates from DAMGO-stimulated MOP-HEK293 and WT-HEK293 cells were combined at different ratios to yield a final volume of 100 μl and analyzed either by western blotting (**j**), pS375-MOP phosphorylation assay (**k**) or np-MOP phosphorylation assay (**l**). Mean of all assay data points were calculated from n = 4 to 5 independent experiments performed in duplicates ± SEM. Where error bars are not apparent, SEM was smaller than symbol size. Western blot images are representatives of at least n = 3 replicates.

divided was integrated into the standard workflow of all subsequent GPCR phosphorylation assays (Fig. 1). Supplementary Fig. 6 shows a schematic representation of different experimental setups for agonist stimulation, antagonist experiments, and inhibitor screening. The agonist setting allows the assessment of concentration–response curves as a percentage of maximal stimulation by endogenous or synthetic agonizts. Finally, we used the 7TM phosphorylation assay to unequivocally detected agonist-induced phosphorylation of endogenous MOP receptors in brain lysates obtained from HA-MOP knock-in mice (Supplementary Fig. 7).

**Ligand profiling in MOP phosphorylation assays.** MOP is a primary target for opioid analgesics. MOP desensitization and tolerance are regulated by phosphorylation of four carboxyl-terminal serine and threonine residues, namely, T370, T376 and T379 in addition to S375[19,20]. To facilitate a detailed assessment of agonist-selective phosphorylation signatures, we generated phosphosite-specific antibodies and validated MOP phosphorylation immunoassay results for each of these sites (Supplementary Fig. 8). We then evaluated a number of chemically diverse

agonizts using the enkephalin derivative DAMGO as standard to calculate concentration–response curves. As depicted in Fig. 3a, b, DAMGO-stimulated phosphorylation with similar efficacy and potency at all four sites (Table 1). DAMGO-induced MOP phosphorylation was inhibited by the antagonist naloxone in a concentration-dependent manner (Fig. 3c). In contrast to the effect of DAMGO, the partial agonist morphine-induced S375 phosphorylation and, with much lower potency, phosphorylation at T370, T379 and T376 (Fig. 3d). The full agonist fentanyl promoted phosphorylation of all four sites with higher efficacy than DAMGO (Fig. 3e). In addition to homologous GRK-mediated phosphorylation, MOP also undergoes heterologous second messenger kinase-mediated phosphorylation, e.g., by protein kinase C (PKC). Notably, when PKC was activated by increasing concentrations of phorbol 12-myristate 13-acetate (PMA), we found potent and selective phosphorylation of T370 (Fig. 3e). Thus, these findings recapitulated those from previous western blot analyses, suggesting that the MOP phosphorylation immunoassay allows rapid and quantitative assessment of agonist-specific phosphorylation patterns[19,20–22]. In addition, antagonist activity was evaluated and homologous and

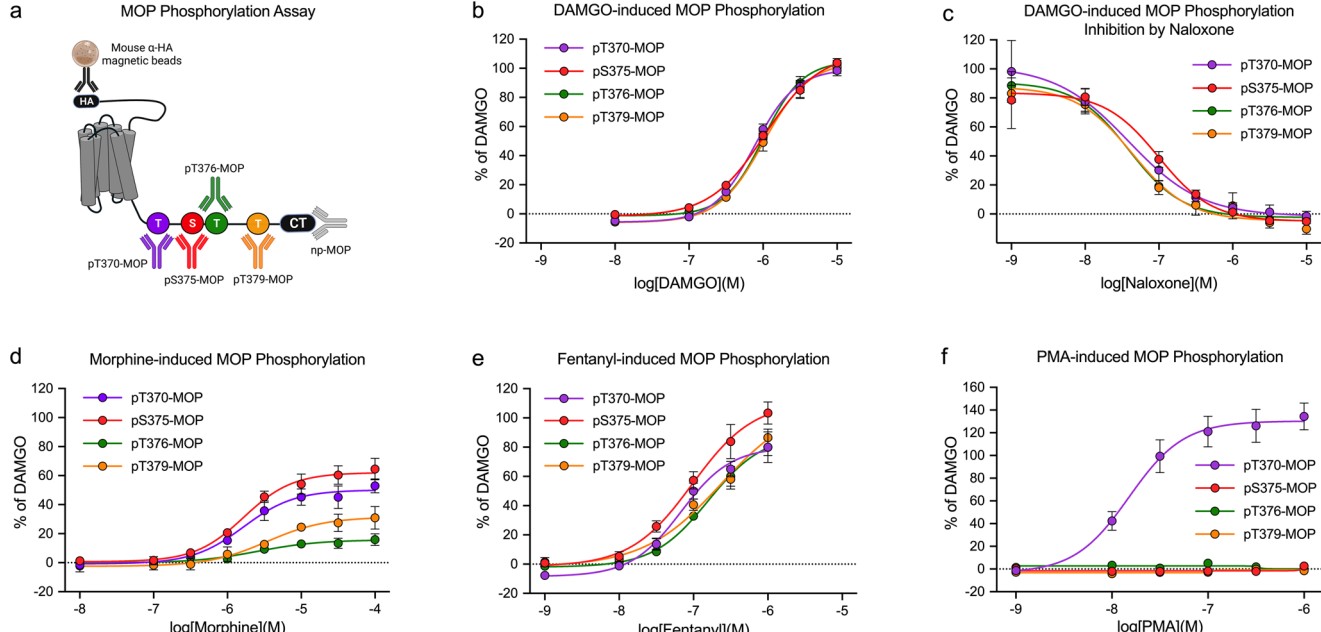

**Fig. 3 Profiling of μ-opioid receptor (MOP) agonizts and antagonists using the 7TM phosphorylation assay. a** Schematic representation showing the MOP binding sites for antibodies against phosphorylated (p)T370-, pS375-, pT376-, and pT379-MOP and MOP independent of phosphorylation (np-MOP). **b–f** Concentration–response curves were generated after treatment with DAMGO (**b**), naloxone (**c**), morphine (**d**), fentanyl (**e**) and the PKC activator phorbol 12-myristate 13-acetate (PMA) (**f**). The cells were exposed to agonizts or PMA for 30 min at 37 °C. The antagonist naloxone was added 30 min prior to DAMGO stimulation. The cells were then lysed and analyzed according to the standard assay protocol. Every graph represents the mean of $n = 5$ independent experiments performed in duplicates ± SEM. Where error bars are not apparent, SEM was smaller than symbol size. All data points were normalized to 10 μM DAMGO stimulation.

**Table 1 Ligand profiling using MOP phosphorylation assays.**

|  | pT370-MOP | | pS375-MOP | | pT376-MOP | | pT379-MOP | |
|---|---|---|---|---|---|---|---|---|
| **Agonist** | $pEC_{50}$ | $E_{max}$ (% DAMGO) | $pEC_{50}$ | $E_{max}$ (% DAMGO) | $pEC_{50}$ | $E_{max}$ (% DAMGO) | $pEC_{50}$ | $E_{max}$ (% DAMGO) |
| DAMGO | 6.1 ± 0.04 | 100 ± 3.2 | 6.0 ± 0.05 | 100 ± 5.3 | 6.0 ± 0.03 | 100 ± 3.5 | 6.0 ± 0.06 | 100 ± 5.9 |
| Morphine | 5.6 ± 0.09 | 52.1 ± 3.9 | 5.8 ± 0.07 | 62.3 ± 3.6 | 5.4 ± 0.17 | 19.2 ± 2.8 | 5.5 ± 0.16 | 37.6 ± 5.6 |
| Fentanyl | 7.2 ± 0.12 | 88.6 ± 7.4 | 7.0 ± 0.14 | 112.8 ± 13.7 | 6.8 ± 0.11 | 90.8 ± 9.9 | 6.6 ± 0.23 | 112.9 ± 9.1 |
| PMA | 7.8 ± 0.12 | 133.9 ± 8.3 | – | – | – | – | – | – |
| **Antagonist** | $pIC_{50}$ | | $pIC_{50}$ | | $pIC_{50}$ | | $pIC_{50}$ | |
| Naloxone | 7.3 ± 0.10 | – | 7.1 ± 0.10 | – | 7.4 ± 0.07 | – | 7.4 ± 0.15 | – |

Data represent means of $n = 5$ independent experiments performed in duplicates ± SE.

**Table 2 Precision of MOP phosphorylation assays.**

| 7TM Phosphorylation Assay | % CV Interassay | % CV Intra-assay | Z′ factor |
|---|---|---|---|
| pT370-MOP | 9.8 | 9.3 | 0.87 |
| pS375-MOP | 15.1 | 5.5 | 0.69 |
| pT376-MOP | 11.9 | 9.5 | 0.80 |
| pT379-MOP | 12.1 | 8.3 | 0.82 |
| np-MOP | 9.0 | 6.5 | N.a.[a] |

Values were calculated from at least 30 determinations performed in duplicate.
[a]Not applicable.

heterologous receptor phosphorylation was differentiated. From this set of measurements, we calculated interassay and intra-assay coefficients of variability (CVs) (Table 2). For the pT370, pS375, pT376, and pT379-MOP phosphorylation assays, the interassay CV was generally at or less than 15%. The intra-assay CV was less than 10%, reflecting the high precision of these assays. In

addition, the calculated Z′ factors for these assays ranged from 0.7 to 0.85, indicating their suitability for high-throughput applications (Table 2). We then performed β-galactosidase complementation assays to study the interrelation between GRK engagement, MOP phosphorylation and arrestin recruitment. In response to DAMGO, GRK2 and GRK3 were recruited with high efficiency, resulting in full phosphorylation of MOP (Fig. 4a, c). In contrast, the morphine-activated receptor recruited GRK2 and GRK3 with much lower potency, which led to only partial MOP phosphorylation (Fig. 4b, d). The concentration–response curves for β-arrestin1 and β-arrestin2 recruitment were virtually identical to those of DAMGO-induced MOP phosphorylation (Fig. 4e, f), whereas morphine-induced MOP phosphorylation was not sufficient to stimulate detectable mobilization of arrestins under these conditions (Fig. 4f, h). When activation of G protein-coupled inwardly rectifying potassium channels (GIRK) was measured using a membrane potential assay, we observed a leftward shift in the concentration–response curves (Fig. 4 and Supplementary Table 1). This leftward shift reflects an amplification mechanism often detected with G protein assays, whereas

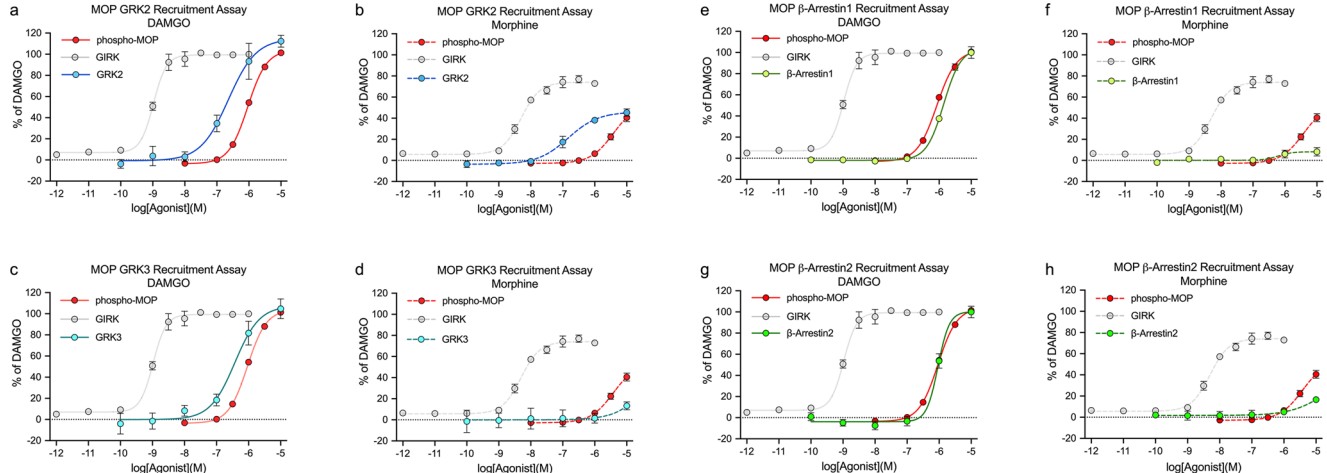

**Fig. 4 Comparison of agonist-induced mouse μ-opioid receptor (MOP) phosphorylation, G protein signaling, and GRK and arrestin binding.** Cells were exposed to increasing concentrations of DAMGO (**a**, **c**, **e**, **g**) or morphine (**b**, **d**, **f**, **h**). Total MOP phosphorylation was determined according to the standard assay protocol and depicted as the mean of pT370-, pS375-, pT376- and pT379-MOP phosphorylation assays (red). G protein signaling was determined using GIRK-mediated changes in membrane potential (gray). Amounts of GRK2 (blue), GRK3 (turquoise), β-arrestin1 (lime) and β-arrestin2 (green) recruited were determined by β-galactosidase complementation assay. Data are means ± SEM from at least $n = 3$ independent experiments performed in duplicates. Where error bars are not apparent, SEM was smaller than symbol size. Note for better comparison to GRK and arrestin receuitment, (**a**, **c**, **e**, **g**) depicted the same data set for DAMGO-induced and (**b**, **d**, **f**, **h**) for morphine-induced GIRK activation (gray). All data points were normalized to 10 μM DAMGO stimulation.

complementation and phosphorylation assays are based on equimolar stoichiometric interactions of the protein partners.

**Identification of GRKs in C5a1 phosphorylation assays.** Next, we tested whether phosphorylation assays can be used for assessing GPCRs, which predominantly undergo GRK5/6-dependent phosphorylation. The complement component 5a receptor 1 (C5a1) is involved in host response to infection and tissue damage. Unlike MOP, C5a1 seems to require primarily GRK5/6 for β-arrestin recruitment[7,23], however, the exact contribution of individual GRK isoforms to C5a1 phosphorylation remains unclear. The generation of phosphosite-specific C5a1 antibodies revealed that at least 8 out of 11S/T residues within the carboxyl-terminal tail undergo agonist-driven phosphorylation. Antibodies targeting three distinct phosphorylation clusters were then extensively characterized and selected for the development of the corresponding pT324/pS327-, pS332/pS334-, pS338/pT339-C5a1 phosphorylation assays (Supplementary Figs. 9–11). We also established two alternative methods for detecting total receptor content to subsequently perform a quantitative phosphorylation analysis (Fig. 5a). First, we generated an anti-nonphospho-(p)-C5a1 antibody (np-C5a1) that detects the receptor in the carboxyl-terminal region in a phosphorylation-independent manner (Fig. 5a). Second, we expressed a receptor construct that contained three consecutive affinity tags in the N-terminus, which facilitated immunoprecipitation using mouse anti-HA antibody magnetic beads and simultaneous detection using rabbit anti-HA antibodies (Fig. 5a). The same receptor construct was then expressed in HEK293 cell clones in which GRK2/3 (ΔGRK2/3-HEK293) or GRK5/6 (ΔGRK5/6-HEK293) expression was knocked out, as well as in the parental HEK293 cell line (Control-HEK293). The cells were stimulated with the synthetic C5a1 agonist C028 (Supplementary Fig. 11). The resulting concentration–response curves showed that for the phosphorylation of proximal sites T324/S327 and S332/S334, the GRK5/6 isoforms were predominantly required, whereas both the GRK2/3 and GRK5/6 isoforms contributed equally to phosphorylation of the distal sites S338/pT339 (Fig. 5b–d and

Supplementary Table 2). These results suggest that 7TM phosphorylation assays can be performed to reveal the differential involvement of individual GRKs in agonist-selective phosphorylation and therefore can provide details in addition to those obtained with conventional β-arrestin recruitment assays.

**GRK inhibitor screening assays.** Screening for selective and membrane-permeable GRK inhibitors is still very challenging. Previous studies have largely relied on measuring receptor internalization or desensitization in identifying cell-permeable GRK inhibitors[24–29]. However, such assays can only indirectly assess GPCR phosphorylation and are difficult to adapt to quantitative high-throughput screening. Therefore, we performed a set of experiments to establish cell-based GRK inhibitor assays. To this end, we capitalized on a recently developed panel of combinatorial HEK293 cell clones in which GRK2/3/5/6 expression was knocked out in various combinations, with one, two, three or all genes knocked out[7]. First, we evaluated MOP phosphorylation through western blot analysis. Phosphorylation was not detected in the quadruple ΔGRK2/3/5/6-HEK293 cells, confirming that DAMGO-induced MOP phosphorylation was entirely mediated by GRKs (Fig. 6a). The results depicted in Fig. 6 a reveal that T376-MOP phosphorylation was virtually absent in the ΔGRK2/3-HEK293 cells, suggesting that T376-MOP phosphorylation in parental Control-HEK293 cells was predominantly mediated by GRK2 and GRK3. In contrast, S375-MOP phosphorylation was clearly detectable in ΔGRK2/3-HEK293 cells, suggesting that residual S375-MOP phosphorylation was mediated by GRK5 and GRK6 (Fig. 6a). These findings were recapitulated with MOP phosphorylation immunoassays, indicating that pT376-MOP phosphorylation in Control-HEK293 cells can be used to screen for GRK2/3 inhibitors (Fig. 6b, c) (Supplementary Fig. 12). Conversely, pS375-MOP phosphorylation in ΔGRK2/3-HEK293 cells can be used to screen for GRK5/6 inhibitors (Fig. 6b, c). Next, we validated this hypothesis using compound101, which is a potent and selective GRK2/3 inhibitor. In fact, compound101 inhibited GRK2/3 activity with a half-maximal inhibitory concentration (IC$_{50}$) of 2.9 μM, but it had no

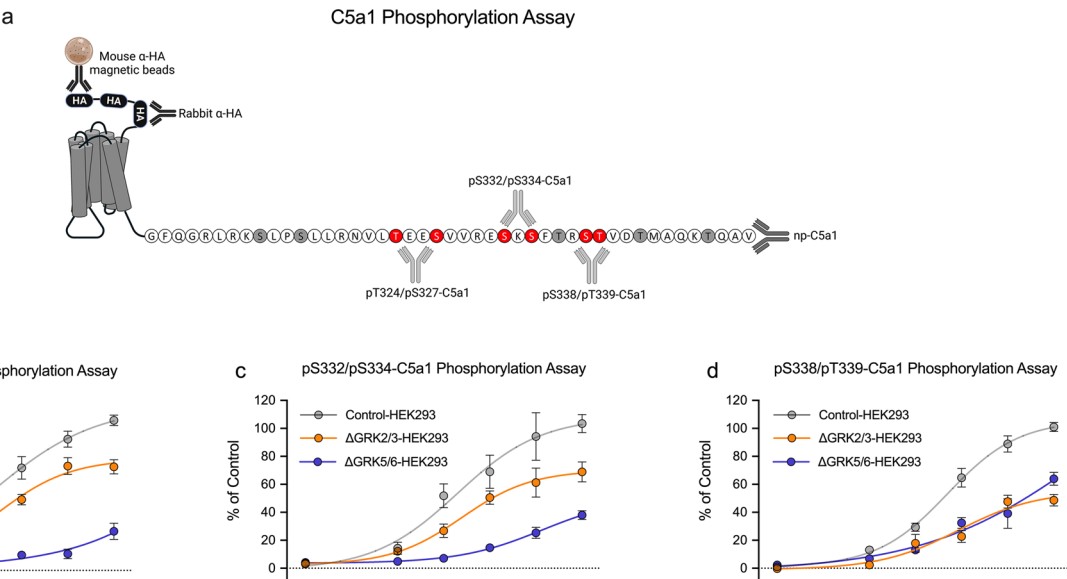

**Fig. 5 Differential involvement of GRK2/3 and GRK5/6 in C5a1 multisite phosphorylation. a** Schematic representation showing C5a1-binding sites for mouse anti-hemagglutinin (HA) magnetic beads, rabbit anti-HA antibodies and antibodies against phosphorylated pT324/pS327-, pS332/pS334-, pS338/pT339-, and C5a1 independent of phosphorylation status (np-C5a1). S/T sites detected using phosphosites-specific C5a1 antibodies are depicted in red. All other potential phosphate acceptor sites are depicted in gray. **b–d** Control-HEK293 cells (gray), ΔGRK2/3-HEK293 cells (orange) and ΔGRK5/6-HEK293 cells (indigo) stably expressing C5a1 were exposed to increasing concentrations of the synthetic agonist C028 and incubated for 30 min at 37 °C. Phosphorylation of T324/S327 (**b**), S332/S334 (**c**) and S338/T339 (**d**) was assessed according to the standard protocol. Means of $n = 5$ independent experiments performed in duplicates ± SEM are presented in concentration–response curves. Where error bars are not apparent, SEM was smaller than symbol size. All data points were normalized to 10 μM C028 stimulation of C5a1-expressing Control-HEK293 cells.

effect on GRK5/6 (Fig. 6d, e and Table 3). We then attempted to identify selective GRK5 inhibitors that display novel chemotypes. To this end, a sublibrary of 16,666 structurally distinct potential kinase inhibitors was tested in vitro in GRK5 kinase assays, resulting in a total of 208 primary hits. We independently confirmed and validated the most potent primary GRK5 hits, followed by hit exploration and medicinal chemistry-based optimization of a genuinely known compound class of ATP competitive kinase inhibitors although not previously described as GRK inhibitors. Hit exploration and the generation of novel analogs quickly led to the identification of more potent and largely double digit nanomolar GRK5 inhibitors as well as to an understanding of the structure-activity-relationship. Ten of such GRK5 inhibitor analogs were selected for further testing in our cell-based GRK inhibitor assays, which led to the discovery of LDC9728, which was 10-fold more effective than compound101 in inhibiting GRK2/3 activity (Fig. 6d and Table 3). LDC9728 was also highly effective in blocking GRK5/6 activity, suggesting that this compound is a highly potent pan-GRK inhibitor (Fig. 6e and Table 3). In addition, we discovered LDC8988 as a potent and selective GRK5/6 inhibitor that has very little effect on GRK2/3 activity (Fig. 6d, e and Table 3 and Supplementary Table 3). These results show that phosphorylation immunoassays can be used to identify relevant GRKs involved in receptor phosphorylation as well as to screen for new cell-permeable GRK inhibitors.

The final experiments were designed to assess the utility of available phosphosite-specific antibodies for the development of phosphorylation assays for other GPCR targets with chemically diverse ligands. In fact, we rapidly established additional assays for the D1 dopamine receptor (Supplementary Fig. 13 and Supplementary Table 4), SST2 somatostatin receptor (Supplementary Fig. 14 and Supplementary Table 4) and CB2

cannabinoid receptor (Supplementary Fig. 15 and Supplementary Table 4), demonstrating the versatility of the 7TM phosphorylation immunoassay.

## Discussion

By combining magnetic bead-based receptor isolation in multiwell plates and using antibodies as phospho-biosensors, we established the 7TM phosphorylation immunoassay. The essential assay components are phosphosite-specific and phosphorylation-independent GPCR antibodies, affinity-tagged receptors, and magnetic beads coated with anti-tag antibodies. The assay is quantitative and particularly effective when phosphorylated and total receptor levels are detected in parallel at the same sample. When phosphorylation-independent anti-GPCR antibodies are not available, receptor constructs with multiple affinity tags and a second affinity tag antibody obtained from another species can be used to measure total receptor levels. The assay is versatile, particularly because it can be established for virtually all 7TMRs provided phosphosite-specific antibodies become available. Given the increased number of available phosphosite-specific GPCR antibodies in recent years[30–38], one can envision that 7TM phosphorylation immunoassays can be established for the majority of pharmacologically relevant GPCRs in the near future.

Our approach is sufficiently sensitive to be performed entirely in a multiwell cell culture format, eliminating the need for technically demanding and time-consuming immunoblot analysis. In fact, the assay is also sensitive enough to detect agonist-induced phosphorylation of endogenously expressed MOP receptors in mouse brain in vivo (Supplementary Fig. 7). Bead-based assays require magnetization during the washing steps. However, laboratory instrumentation for the automated processing of bead-based immunoassays is readily available. The

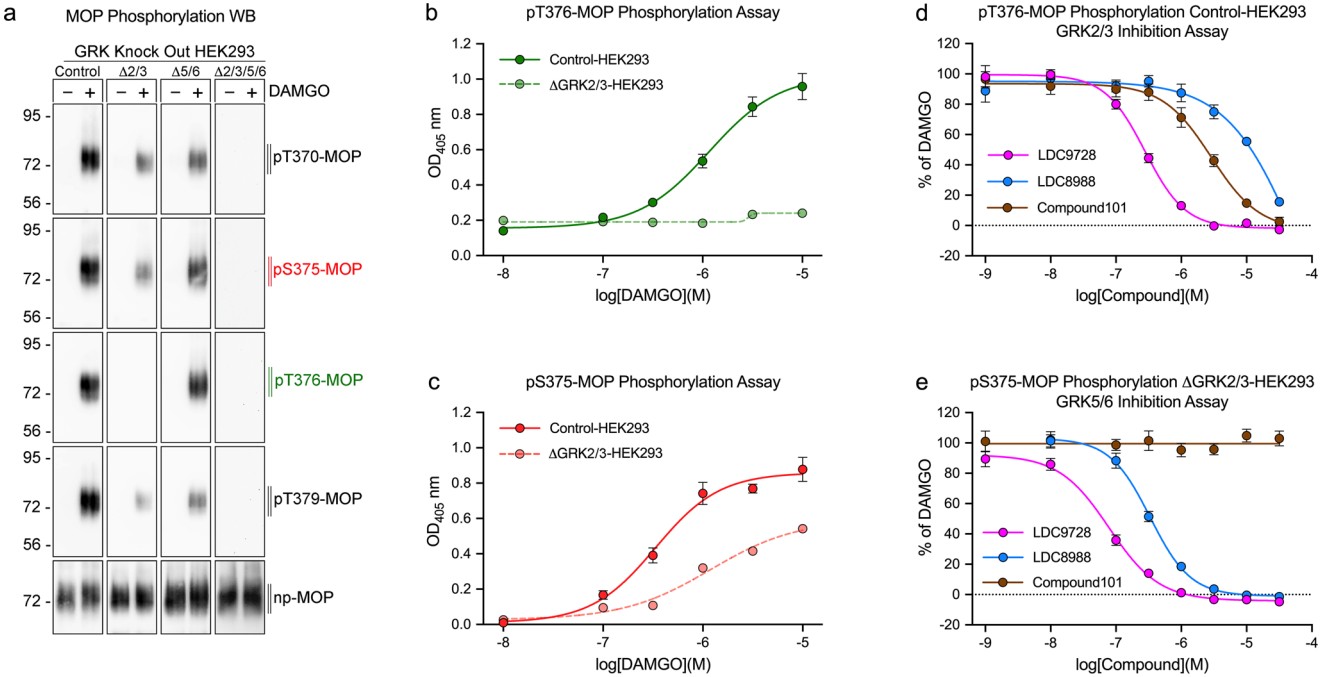

**Fig. 6 GRK inhibitor screening using the 7TM phosphorylation assay. a** Control-HEK293, ΔGRK2/3-HEK293, ΔGRK5/6-HEK293 and ΔGRK2/3/5/6-HEK293 cells stably expressing mouse μ-opioid receptor (MOP) were either unstimulated (−) or exposed to 10 μM DAMGO (+) for 30 min at 37 °C. Subsequently, T370, S375, T376 and T379 phosphorylation was determined by western blot analysis. **b** MOP-expressing Control-HEK293 and ΔGRK2/3-HEK293 cells were exposed to increasing concentrations of DAMGO, and T376 phosphorylation was determined according to the standard protocol. **c** MOP-expressing Control-HEK293 and ΔGRK2/3-HEK293 cells were exposed to increasing concentrations of DAMGO, and the degree of S375 phosphorylation was determined. **d** MOP-expressing Control-HEK293 cells were treated with increasing concentrations of the GRK inhibitors LDC9728, LDC8988 or compound101 for 30 min prior to stimulation with 10 μM DAMGO, and T376 phosphorylation was then determined. **e** MOP-expressing ΔGRK2/3-HEK293 cells were treated with increasing concentrations of the GRK inhibitors LDC9728, LDC8988 or compound101 for 30 min prior to stimulation with 10 μM DAMGO, and S375 phosphorylation was then determined. The data points shown in (**b**, **c**) represent optical density read at 405 nm ($OD_{405}$). Data points shown in (**b**, **c**) represent optical density readings at 405 nm ($OD_{405}$). Data points shown in (**d**) were normalized to 10 μM DAMGO stimulation of MOP-expressing Control-HEK293 cells. The data points shown in (**e**) were normalized to 10 μM DAMGO stimulation of MOP-expressing ΔGRK2/3-HEK293 cells. Western blot images are representatives of $n = 5$ independent experiments. Immunoassay data are means ± SEM of $n = 5$ independent experiments performed in quadruplicates. Where error bars are not apparent, SEM was smaller than symbol size.

**Table 3 GRK inhibitor profiling by 7TM phosphorylation assay.**

| Inhibitor | GRK2/3 Inhibition Assay IC$_{50}$ (μM) | GRK5/6 Inhibition Assay IC$_{50}$ (μM) |
|---|---|---|
| LDC9728 | 0.28 ± 0.03 | 0.08 ± 0.04 |
| LDC8988 | 66.3 ± 1.29 | 0.33 ± 0.04 |
| Compound101 | 2.87 ± 0.09 | – |

Data represent the mean of $n = 5$ independent experiments performed in quadruplicates ± SE.

calculated Z' factors for our 7TM phosphorylation assays indicate the suitability of the assay for use in large-scale, high-throughput screening. Magnetic bead-assisted immunocapture and subsequent quantification of isolated targets has been described for other phosphoproteins[39] and the 7TM phosphorylation immunoassay expands this concept now to GPCRs. By using fluorescently-labeled beads, it should be possible to establish multiplex applications for quantitative assessment of GPCR phosphorylation patterns in the same sample.

Phosphosite-specific anti-GPCR antibodies can be used as biosensors to detect receptor activation, helping to distinguish agoniztic and antagonistic properties when profiling new ligands[40]. In principle, this assay can also be used for screening orphan GPCRs when the precise G protein-coupling and second messenger pathways are unknown. For a detailed analysis, concentration–response curves can be constructed to distinguish partial and full agonizts[41]. GPCRs can adopt different agonist-dependent conformations that favor one signaling pathway over another, a phenomenon termed functional selectivity or biased signaling[42,43]. Determination of ligand bias is often based on analysis of G protein signaling and arrestin binding. Both the active receptor conformation and the phosphorylation of site-specific clusters of hydroxyl amino acid residues contribute to efficient arrestin recruitment, and for many receptors, GRK overexpression is required to detect arrestin binding in response to partial agonizts. Therefore, a detailed assessment of agonist-driven phosphorylation patterns can provide valuable insights into ligand pharmacology in the presence of endogenous GRK levels[44].

Agonist-driven homologous phosphorylation is mediated by GRKs, while heterologous phosphorylation involves second messenger-regulated kinases such as protein kinase A or protein kinase C. By designing appropriate assay conditions and applying chemical inhibitors, our approach can be used to discriminate between different phosphorylation modes and help identify relevant kinases. However, to date only few potent and selective GRK inhibitors have been discovered[24–29,45]. Among these compound101 is most frequently used to assess the involvement of GRK2/3. We created GRK2-/3-/5-/6-knockout cells, including double, triple and quadruple knockout cells[7], which enabled us to evaluate the precise contribution of individual GRKs to site-specific phosphorylation. We then extended this approach to

develop cell-based screening assays for membrane-permeable GRK inhibitors that led to the rapid identification of LDC8988, a selective GRK5/6 inhibitor, and LDC9728, a highly potent pan-GRK inhibitor. Both compounds belong to a novel class of GRK inhibitors that is more potent and structurally distinct from currently available inhibitors. Such inhibitors are much-needed tool compounds for elucidating the physiological functions of GRKs.

In summary, we provide proof of concept for a first-in-class phosphorylation immunoassay that can be easily adapted to a variety of prototypical GPCRs as well as atypical 7TMRs. This approach is not limited to agonist profiling and identification of kinases involved in receptor phosphorylation but can be extended to GRK inhibitor screening and orphan receptor ligand pairing. Finally, the 7TM phosphorylation assay is fast, robust and reliable, thus fulfilling all the requirements for extensive application in academic and pharmaceutical research.

## Methods

**Antibodies, cells and reagents**. The phosphosite-specific MOP antibodies against pT370-MOP (7TM0319B), pS375-MOP (7TM0319C), pT376-MOP (7TM0319D), and pT379-MOP (7TM0319E); the phosphosite-specific C5a1 antibodies against pT324/pS327-C5a1 (7TM0032A), pS332/pS334-C5a1 (7TM0032B), and pS338/pT339-C5a1 (7TM0032D); and phosphosite-specific D1 dopamine receptor antibodies against pT354-D1 (7TM0214A) and pS372/pS373-D1 (7TM0214B); and phosphosite-specific SST2 antibodies against pS341/pS343-SST2 (7TM0356A) and pT356/pT359-SST2 (7TM0356C); and phosphosite-specific CB2 cannabinoid receptor antibodies pS335/pS336-CB2 (7TM0057A) and pS338/pS340-CB2 (7TM0057B); as well as the phosphorylation-independent antibodies against np-MOP (7TM0319N), np-C5a1 (7TM0032N), np-SST2 (7TM0356N), and rabbit polyclonal anti-HA antibodies (7TM000HA) were provided by 7TM Antibodies (Jena, Germany). Phosphosite-specific antibodies were affinity-purified against their immunizing phosphorylated peptides and subsequently cross-adsorbed against the corresponding unphosphorylated peptides. All phosphosite-specific antibodies have been extensively characterized using western blot and dot blot analyses before application in 7TM phosphorylation assays[7,16,19–22,34,35,46–49], (Supplementary Figs. 9, 10, Supplementary Table 7) (www.7tmantibodies.com). HEK293 cells were originally obtained from DSMZ Germany (ACC 305). Stable MOP-, C5a1- or SST2-expressing cells and GRK-knockout cells were generated as previously described. The plasmid encoding murine MOP with one amino-terminal HA-tag was custom-synthesized by imaGenes (Berlin, Germany). The plasmid encoding human CB2 with three amino-terminal HA-tags was custom-synthesized by GenScript (Rijswijk, Netherlands). The plasmids encoding human C5a1, D1, SST2 with three amino-terminal HA-tags were obtained from the cDNA resource center (C5R010TN00, DRD010TN00, SSTR20TN00) (www.cDNA.org). The GRK inhibitors LDC9728 and LDC8988 were provided by the Lead Discovery Center (LDC, Dortmund, Germany). The chemical structures for LDC9728 and LDC8988 will be reported in a separate publication.

**Cell culture**. HEK293 cells were cultured in Dulbecco's modified Eagle's medium (DMEM; Capricorn Scientific, DMEM-HXA) supplemented with 10% fetal calf serum (FCS, Capricorn Scientific FBS-11A) and a 1% penicillin and streptomycin mixture (Capricorn Scientific PS-B) at 37 °C with 5% $CO_2$. Cells were passaged every 3–4 days and regularly checked for mycoplasma infections using a GoTaq G2 Hot Start Taq Polymerase kit from Promega.

**7TM phosphorylation assay**. The 7TM phosphorylation assay was performed according to the protocol outlined in Fig. 1 with either wild-type (Control-HEK293) or GRK-knockout HEK293 cells stably transfected with MOP or C5a1. The cells were seeded into 96-F-bottom-well cell culture microplates (Greiner Bio-One 655180) coated with 0.2 μg/ml poly-L-lysine (Sigma-Aldrich P1274) at a density of 80,000–100,000 cells per well and grown overnight to 95% confluency. The cells were then stimulated with the respective agonizts for 30 min at 37 °C. MOP was stimulated with DAMGO (Sigma-Aldrich E7384, in water), morphine (Hameln 03763738, in water), fentanyl (B. Braun 06900650, in water) or PMA (Tocris 1201, in dimethyl sulfoxide (DMSO)). C5a1 was stimulated with the FKP-(D-Cha)-Cha-r peptide agonist C028 (AnaSpec AS65121, in water). In antagonist and GRK inhibitor experiments, naloxone (Ratiopharm 04788930, in water), compound101 (Hello Bio HB2840, in DMSO), LDC9728 (Lead Discovery, in DMSO) or LDC8988 (Lead Discovery, in DMSO) was added 30 min before DAMGO stimulation. After washing with PBS, the cells were lysed in 150 μl/well detergent buffer (150 mM NaCl; 50 mM Tris-HCl, pH 7.4; 5 mM EDTA; 1% Igepal CA-360; 0.5% deoxycholic acid; and 0.1% SDS) supplemented with protease and phosphatase inhibitor cocktails (Roche #04693132001 and #04906845001, respectively) on an orbital shaker for 30 min at 4 °C. In dephosphorylation experiments,

phosphatase inhibitors were omitted from the indicated samples. Lysates were cleared by centrifugation at $3700 \times g$ for 20 min at 4 °C using a microplate centrifuge (Thermo Fisher Multifuge X Pro). The supernatants were then transferred into 96-U-bottom-well assay plates (Greiner Bio-One 650101). Each sample was divided and placed into two corresponding wells of the assay plate for parallel detection of phosphorylated (80 μl of lysate per well) and total (40 μl of lysate per well) receptor levels according to the scheme depicted in Supplementary Fig. 6. For in vivo phosphorylation experiments, knock-in mice expressing HA-MOP (Oprm1em1Shlz, MGI:6117675) were treated and brain lysates were prepared as described[16]. Mouse anti-HA magnetic beads (2 μl/well) (Thermo Fisher 88837) were washed in detergent buffer, diluted and vortexed for more accurate pipetting. A total volume of 40 μl of the bead slurry, which contained 20 μg magnetic beads with a diameter of 1 μm, was then added to each well. Assay plates were incubated on a microplate shaker (Corning LSE, digital) at 500–700 rpm for 2 h at 4 °C. The beads were washed three times with 150 μl of PBS with 0.1% Tween 20 (PBS-T) under magnetic force. In the initial experiments, magnetic force was applied by placing the assay plates on a handheld magnetic separation block (V&P Scientific 771HH-H/Millipore 40-285). In all subsequent experiments, an automated microplate washer (BioTek 405™ TS Microplate Washer) was used. After the first washing cycle, primary rabbit phosphosite-specific and phosphorylation-independent antibodies were added at a final concentration of 3–5 μg/ml in 60 μl of PBS-T. The plates were then incubated either for 2 h at room temperature or overnight at 4 °C on a microplate shaker. In peptide neutralization experiments, phosphosite-specific antibodies were incubated with 1 μg/ml phosphorylated peptide used for immunizations or the corresponding unphosphorylated peptide for 1 h at room temperature on a turning wheel. After the second washing cycle, anti-rabbit horseradish peroxide (HRP)-linked secondary antibody (Cell Signaling Technology #7074) was added to PBS-T to a final dilution of 1:300 and incubated on a microplate shaker for 2 h at room temperature. After the third washing cycle, 100 μl of Super AquaBlue detection solution (Thermo Fisher 00-4203-58), an enhanced ABTS substrate solution, was added to each well, and the plates were incubated for 2–8 min until an OD at 405 nm ($OD_{405}$) between 1.0 and 1.4. The color reaction was stopped by the addition of 100 μl of 0.625 M oxalic acid. The assay plates were then placed on a magnetic separation block, and 150 μl of the supernatant was transferred to a transparent 96-well F-bottom detection plate (Greiner Bio-One 655182). The $OD_{405}$ was then determined using a FlexStation 3 microplate reader (Molecular Devices). Data were acquired with SoftMax Pro 5.4 software and analyzed with Excel 16.0 software. First, the mean of all background controls (without primary antibodies) was subtracted from all values. To normalize the signal intensity, the phosphorylation signal was multiplied by the quotient of the mean of all loading controls divided by the respective loading control. Therefore, the result was adjusted to the amount of receptor for each corresponding sample. This method provides the required information to create quantitative concentration–response curves based on raw $OD_{405}$ values. In each plate, samples of the control agonist were included to facilitate calculation of the results as a percent of agonist control (Supplementary Fig. 1). Values were displayed as concentration–response-curves of at least five independent experiments performed in duplicates and generated with GraphPad Prism 9.3.1 software.

**Western blot analysis**. For dephosphorylation and detection limit experiments, HEK293 cells were grown, treated and processed as described for the 7TM phosphorylation assay until the first wash cycle of the magnetic beads was completed. Subsequently, 50 μl of 1× SDS-sample buffer (100 mM DTT, 62.5 mM Tris-HCl, 20% glycerol, 2% SDS, and 0.005% bromophenol blue) was added to each well, and the plate was incubated for 25 min at 43 °C. The assay plates were then placed on a magnetic separation block, and the supernatant was loaded onto an 8% poly-acrylamide gel for immunoblot analysis (15 μl per lane). For all other experiments, cells were seeded in 60-mm dishes (Greiner Bio-One 628160) coated with 0.1 μg/ml poly-L-lysine (Sigma-Aldrich P1274) and grown to 90% confluency. The cells were then stimulated with an agonist for 30 min at 37 °C. After washing with PBS, the cells were lysed in 800 μl of detergent buffer (150 mM NaCl; 50 mM Tris-HCl, pH 7.4; 5 mM EDTA; 1% Igepal CA-360; 0.5% deoxycholic acid; and 0.1% SDS) supplemented with protease and phosphatase inhibitor cocktails (Roche #04693132001 and #04906845001, respectively). After centrifugation at $14,000 \times g$ for 30 min at 4 °C, lysates were incubated with 40 μl of mouse anti-HA agarose beads (Thermo Fisher 26182) at 4 °C on a turning wheel for 2 h. The beads were then washed three times with detergent buffer, 60 μl of 1× SDS-sample buffer was added, and the samples were heated to 43 °C for 25 min to elute receptors from the beads. Supernatants were loaded onto 8% polyacrylamide gels. After gel electrophoresis, samples were blotted onto a PVDF membrane using a semidry electroblotting system. Phosphorylation was detected by incubating rabbit phosphosite-specific antibodies at concentrations of 1–2 μg/ml in 5% bovine serum albumin (BSA)/TBS overnight at 4 °C. Signals were visualized with anti-rabbit HRP-linked secondary antibodies (Cell Signaling Technology #7074) and a chemiluminescence detection system (Thermo Fisher 34075). The blots were subsequently stripped and reprobed with a rabbit phosphorylation-independent antibody or rabbit anti-HA-tagged antibody to ensure equal loading of the gels. Images depict a representative of at least $n = 3$ independent replicates. Western blot signals were imaged and quantified using a Fusion FX7 imaging system (Peqlab).

**Arrestin and GRK binding assays**. Agonist-dependent binding of GRK2/3 and arrestin1/2 to a MOP was determined using a β-galactosidase complementation assay as previously described[47]. HEK293 cells stably expressing MOP in which the C-terminal was fused with a β-Gal enzyme fragment (β-Gal1–44) and stably or transiently expressing β-arrestin1/2 or GRK2/3 fused to an N-terminal deletion mutant of β-Gal (β-Gal45–1043) were used. Receptor activation resulted in complementation of β-Gal fragments that generated an active enzyme. Thus, the enzyme activation levels are a direct result of MOP activation and are quantitated using a chemiluminescent β-Gal substrate (PJK Biotech #103312). Cells were plated in 48-well plates and grown for 48 h. After 60 min of agonist exposure, a cell lysis reagent was added, and luminescence was recorded with a FlexStation 3 microplate reader (Molecular Devices).

**G protein-gated inwardly rectifying potassium channel assay**. To analyze $G_i$ signaling, changes in membrane potential produced by activation of G protein-gated inwardly rectifying potassium channel (GIRK) were measured as previously described[50,51]. HEK293 cells were stably transfected with HA-MOP and GFP-conjugated GIRK2 channel plasmids (OriGene). The cells were then seeded in 96-well plates and allowed to grow at 37 °C in 5% $CO_2$ for 48 h. Hank's balanced salt solution (HBSS) with 20 mM HEPES solution (1.3 mM $CaCl_2$, 5.4 mM KCl, 0.4 mM $K_2HPO_4$, 0.5 mM $MgCl_2$, 0.4 mM $MgSO_4$, 136.9 mM NaCl, 0.3 mM $Na_2HPO_4$, 4.2 mM $NaHCO_3$ and 5.5 mM glucose; pH 7.4) was used to wash the cells. A membrane potential dye (FLIPR Membrane Potential kit BLUE, Molecular Devices R8034) was reconstituted according to the manufacturer's instructions. To each well, 90 μl of the HBSS/HEPES buffer solution and an equal volume of the membrane potential dye were added to a final volume of 180 μl per well. The cells were then incubated at 37 °C for 45 min. Test compounds were prepared in buffer solution containing HBSS and 20 mM HEPES solution (pH 7.4) at tenfold the final concentration to be measured. Fluorescence measurements were performed with a FlexStation 3 microplate reader (Molecular Devices) at 37 °C with excitation at 530 nm and emission at 565 nm. Baseline readings were taken every 1.8 s for 1 min. After 60 s, 20 μl of the test or vehicle control was injected into each well containing cells incubated with the dye to a final in-well volume of 200 μl, which resulted in a 1:10 dilution of the test compound. The change in dye fluorescence was recorded for 240 s with SoftMax Pro 5.4 software. Peak fluorescence values were obtained after subtraction of baseline readings for each sample and then used to calculate concentration–response curves using Origin.

**Saturation binding assay**. Binding experiments were performed on membranes prepared from wild-type and HA-MOP-transfected HEK293 cells using [³H] DAMGO (51.7 Ci/mmol) (Perkin Elmer NET902250UC) as described previously[52]. Briefly, cells grown to confluency were harvested in PBS and stored at −80 °C. Saturation binding experiments were performed with 50 mM Tris-HCl buffer (pH 7.4) in a final volume of 1 ml containing 30–40 μg of membrane protein. Membranes were incubated with different concentrations of [³H]DAMGO (0.05–9 nM) at 25 °C for 60 min. Nonspecific binding was determined in the presence of 10 μM unlabeled DAMGO. Reactions were terminated by rapid filtration through Whatman glass GF/C fiber filters. The filters were washed three times with 5 ml of ice-cold 50 mM Tris-HCl buffer (pH 7.4) with a Brandel M24R cell harvester. Bound radioactivity retained on the filters was measured by liquid scintillation counting using a Beckman-Coulter LS6500. All experiments were repeated three times with independently prepared samples. Nonlinear regression analysis of the saturation binding curves was performed with GraphPad Prism.

**In vitro GRK2 and GRK5 kinase assays**. The kinase inhibitory activity of LDC8988 and LDC9728 against GRK2 and GRK5 was assessed using the Lance kinase activity assay, which is based on the detection of a phosphorylated Ulight-peptide substrate by a specific europium-labeled anti-phospho peptide antibody. Binding of a kinase inhibitor prevents phosphorylation of the Ulight-substrate resulting in loss of the FRET signal. The assay was performed at an ATP concentration representing the $K_m$ of ATP for GRK2 and GRK5 (Supplementary Tables 5, 6). For $IC_{50}$ determination, 4 μl kinase working solution in assay buffer (50 mM HEPES pH 7.5, 10 mM $MgCl_2$, 1 mM EGTA, 0,01% Tween20, 1% DMSO, 2 mM DTT) and 4 μl of substrate working solution in assay buffer were transferred into assay plates (Corning #4513). Compound was added via an Echo acoustic dispenser (BeckmanCoulter) in a concentration range from 10 to 0.0025 μM. Reaction was started by addition of 2 μl ATP. After 1 h incubation at room temperature, the reaction was stopped with 10 μl detection mix containing the anti-phospho peptide antibody and 10 mM EDTA. After a second incubation period of 1 h at room temperature the FRET signal was measured at 340 nm excitation, 665 nm and 615 nm emission with an Envision spectrophotometer (Perkin Elmer) with 50 μs delay and 300 μs integration time. $IC_{50}$ values were calculated with concentration–response-curves with Quattro Workflow.

**Statistics and reproducibility**. All assays were performed in duplicates and repeated in least five independent experiments. Concentration–response-curves were generated with GraphPad Prism 9.3.1 software. Western blot assays were repeated at least three times. Western blot signals were imaged and quantified using a Fusion FX7 imaging system (Peqlab) and quantified using ImageJ.

**Reporting summary**. Further information on research design is available in the Nature Research Reporting Summary linked to this article.

## Data availability

All data supporting the findings of this study are available within the article and its Supplementary Information files (Supplementary Data 1–3). Additional information, relevant data and unique biological materials will be available from the corresponding author upon reasonable request.

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

## Acknowledgements

We thank Ulrike Schiemenz, Svetlana Würl, and Monique Brendel for expert technical assistance. This work was supported by the European Regional Development Fund (EFRE) and the Free State of Thuringia (grant: 2020 FE 0146) to 7TM Antibodies GmbH. J.K. received a fellowship from 7TM Antibodies and the Free State of Thuringia. Screening and compound synthesis at the Lead Discovery Center GmbH was financially supported by Prof. Dr. Axel Ullrich in his role as a Director of the Max-Planck Institute for Biochemistry, Martinsried, Germany.

## Author contributions

S.S. conceived and initiated the project and designed all experiments with J.K. and N.K.B. N.K.B., A.S., and J.K. performed and analyzed the phosphorylation assays. F.N. contributed to design and generation of phosphosite-specific antibodies and performed affinity purification of all antibodies used in this study. J.D. and C.H. engineered and provided all GRK-knockout cells. N.K.B., S.F. and A.K. performed and analyzed the immunoblot experiments. E.M.-T. performed the GRK and arrestin recruitment assays. P.D. performed the membrane potential assays. C.D., J.E., J.E.E., M.B., and B.K. synthesized and characterized novel GRK inhibitors. C.S. and H.M performed peptide arrays. M.G. and M.S. performed saturation binding experiments. The paper was written and revised by S.S., J.K., N.K.B., and R.K.R. with input from the other authors.

## Funding

## Competing interests

S.S. is the founder and scientific advisor of 7TM Antibodies GmbH, Jena, Germany. S.S. and J.K. are inventors of 7TM Antibodies GmbH patent EP22167875 on this work. F.N. is an employee of 7TM Antibodies. C.D., J.E., J.E.E., M.B. and B.K. are all employees of the Lead Discovery Center GmbH without shareholding. All other authors declare no competing interests.
