## [Peer Review File · Communications Biology]

Reviewers' comments:

Reviewer #1 (Remarks to the Author):

This manuscript by Kaufmann et al. describes the development and validation of a 96 well plate-based immunoassay for detecting and quantifying the phosphorylation of G protein-coupled receptors. It is primarily heterologous in nature, utilizing immunoprecipitation of transfected, epitope-tagged receptors and probing these immunoprecipitates with antibodies raised against phosphopeptides encompassing known phosphorylation sites on the receptor. The manuscript details carefully controlled experiments and thorough validation of the receptor harvesting and detection protocols. The authors successfully applied this approach to the μ -opioid receptor and then to the C5a1 receptor and were able to detect multi-site phosphorylation in both receptors in response to various agonists – moreover, they were able to define the GRKs involved and observed some correlation to the association with β -arrestins. Finally, the authors detailed the utility of this approach to successfully screen for small molecule GRK inhibitors. Overall, this approach is a useful addition to the GPCR signalling/screening armoury and is certainly a useful adjunct/advance over traditional ^{32}P -labelling. My comments are as follows:

- 1) One obvious limitation of the approach is that the technology is completely dependent on the availability and specificity of phosphorylation-state specific antibodies. This limitation should be explicitly stated in the discussion and some balanced comments added regarding this important challenge as well as some other points raised below.
- 2) The development of such phosphorylation-state specific antibodies cannot be assumed a priori, and many (most) phosphorylation sites have not been empirically determined. Some have been presumed indirectly by mutagenesis and even less is known about the hierarchy and co-operativity of multi-site phosphorylation events. Some recognition of this fact as a major, likely barrier to the broad uptake of this technique should be acknowledged.
- 3) In many GPCRs, there are clusters of closely adjacent serines/threonines and it is not clear from the work presented as to whether there is a likelihood that separate antibodies targeting closely position phospho-sites would sterically hinder and compete with each other in this assay if it were to be multiplexed.
- 4) The level of heterologous expression was high (300,000 receptors per cell; ~ 2000 fmol/mg protein) and so some comment on the applicability/sensitivity or otherwise of the technique to eventually be applied to endogenous receptor phosphorylation is warranted (availability of antibody against native receptor notwithstanding).

Reviewer #2 (Remarks to the Author):

Kaufman et al. describe a new cell-based immunoassay approach to evaluate the ability of the various GRKs to install distinct phosphorylation sites in 7TM receptors, and, based on the robustness of the assay, to screen for novel GRK inhibitors based on the ability of ligands to interfere with phosphorylation at these sites.

The key piece of innovation here is their ability to exploit phosphosite-specific antibodies acting on various GPCRs. They then could use a plate-based immunoassay to detect which sites are labeled under various experimental conditions, and also successfully used the assay to identify which GRKs phosphorylate each specific site in the MOR and other receptors in a quantitative way that cannot be achieved, for example, by phosphoproteomics. The paper is generally well written and figures and tables are for the most part very well put together and easy to interpret.

Enthusiasm is however dampened by the facts that 1) there are far easier and straightforward ways to screen for selective GRK inhibitors, and these have been described now for quite a few years 2) the insular writeup that does not compare and contrast this work with the efforts and products of other

labs in developing GRK inhibitors, and 3) the fact that at the end of the day, it is hard to know how much of what was learned about GRK vs. site applies to real physiological processes because these immunoassays are performed in engineered HEK cells overexpressing receptors. We don't know what happens or what GRK has priority in most cell types where the protein concentrations are much lower and not all kinases are equally expressed or available locally. One may indeed use this assay to discover what kinase phosphorylates a GPCR under these experimental conditions, but only at the first site (usually many more sites are modified in a receptor to drive arrestin recruitment). And so this kind of data can only give one a lead for what to look for in vivo.

Editorial comments (Major)

1) One thing that makes a screening assay powerful/more notable is its generalizability. It is true the authors/7TM Antibodies GmbH have developed Abs for certain phosphosites in various GPCRs and tested them here. But for how many GPCRs is this possible? Also, although they were affinity purified against immunizing phosphopeptides, there is no characterization here of these Abs here in terms of their selectivity and affinity. Historically, GPCR phospho-specific Abs have been plagued by these issues.

2) The significance is diminished because, unlike the author's claim, they are not first to identify selective and permeable inhibitors for GRKs. Selective GRKs that have been reported by multiple group, but the authors do not describe these other studies. This leads to some apparent over sell: "Next, we validated this hypothesis using compound 101, which is the most potent and selective GRK2/3 inhibitor available to date" (page 12...the reference for this compound was not cited by the way: <https://pubmed.ncbi.nlm.nih.gov/21596927/> and <https://pubmed.ncbi.nlm.nih.gov/28699740/>). The problem with this statement is that other groups generated similarly potent and equally selective compounds (at least <https://pubmed.ncbi.nlm.nih.gov/33038544/>;<https://pubmed.ncbi.nlm.nih.gov/28323425/>;<https://pubmed.ncbi.nlm.nih.gov/32234810/>). These compounds seem to be effective in cells as well.

Thus one does not need this particular assay, despite its benefits, to find selective and cell permeable GRK2 inhibitors.

Also on page 12: "In addition, we discovered LDC8988 as the first potent and selective GRK5/6 inhibitor that has very little effect on GRK2/3 activity". This is also not true. Several groups have also reported potent GRK5 inhibitors that are also highly selective and very likely cell permeable (<https://pubmed.ncbi.nlm.nih.gov/34291633/>;
<https://pubmed.ncbi.nlm.nih.gov/33393767/>)

The real strength of the approach described herein is instead to assess the ability of each kinase to phosphorylate each individual site in a GPCR when they have the appropriate Abs. The weakness is that it will not be able to tell you who phosphorylates the second site/cluster after the first or what goes on in a physiologically relevant situation. One site does not make a barcode.

3) Supplementary Table 6 (the Lance kinase assay) seems to show n=1 data which is not allowable. This is important data actually because it shows that the compounds identified are at least hitting the expected target.

4) Although to be described in another paper, we know nothing about the selectivity (kinome or otherwise) of the identified compounds, which makes is a little hard to get excited.

Editorial comments (Minor)

1) Page 11. The authors state that "7TM phosphorylation assays can be performed to reveal the differential involvement of individual GRKs in generating agonist-selective phosphorylation barcodes".

This is slight oversell, because one or even two phosphosites does not beget a barcode.

2) "Western" should not be capitalized

3) The standard in pharmacology is to report data in terms of SD, not SEM

4) Where possible, use scatterplots for the bar graph data.

5) Please consider using consistent and reasonable number of significant figures for data. Sometimes 1 is used, sometimes 2, 3, or 4. Or even 5. See Supplementary Table 1 for an example of the full range.

6) Page 11. "Screening for selective and membrane-permeable GRK inhibitors is still very challenging." This actually is not true. There are quite a few other reports now of selective GRK inhibitors that are membrane permeable.

7) Page 15. " To date, only the involvement of GRK2/3 can be determined using compound101.... These novel inhibitors are much-needed tools for elucidating the physiological functions of GRKs." The first statement is not true (see above comments) and the last comment seems like oversell because other labs have developed compounds that are obviously working in vivo.

8) The "linear" curves fit in Supplementary Figure 4 look more like they are saturating. I am not sure one can believe the fits as defined over the range of data selected.

9) The DAMGO and morphine GIRK data shown in Fig. 4 is duplicated 4 times among the panels. The caption needs to explicitly state this. This way of presenting the GIRK data does not seem entirely reasonable to me as it implies it was used as a control in each set of assays. Maybe it was.

10) Delete "gold" on page 9. The following sentence also also needs another "and" : "In addition, antagonist activity was evaluated, homologous and heterologous receptor phosphorylation was differentiated

11) Typo, page 8. "we used the 7TM phosphorylation assay to unequivocally detected"

Comment

The data shown in Supplementary figure 11 is odd because it is essentially a step function. I don't think there could be any feed-forward action here to create such a scenario. Do the authors have an explanation? Hard to know what to make of this data because it is clearly the odd man out for all the GPCRs assessed. Without a good explanation, one might want to omit.

Reviewer #3 (Remarks to the Author):

The manuscript by Kaufmann et al. describes a method for quantitative evaluation of GPCR phosphorylation based on immunoprecipitation and magnetic bead separation. Firstly, they used magnetic beads to immunoprecipitate the affinity-tagged receptors and then detected the protein using phosphorylation-state-specific and phosphorylation-independent anti-GPCR antibodies. In addition, they demonstrated this concept by treating the two GPCRs with different agonists or antagonists, and generating concentration-response curves. Finally, they established an assay for selective cellular GPCR kinase (GRK) inhibitors, which enabled rapid identification of selective GRK5/6 inhibitors (LDC8988) and high-efficiency PAN GRK inhibitors (LDC9728). This technique can be used as one of the methods to study phosphorylation by GPCR. This study is very interesting and provides

new insights into GPCR phosphorylation analysis, which can be widely used for ligand analysis and inhibitor screening. However, there are still some puzzles for the author to solve.

1.The author mentioned several times in this manuscript that a Bead-based GPCR phosphorylation immunoassay can quantitatively analyze the phosphorylation of GPCR, but there seems to be no sufficient evidence to explain this in the manuscript. In the case of quantitative analysis, it should be specified to what extent the technique can pinpoint the magnitude of phosphorylation modification.

2.As we all know, there are more than 800 kinds of GPCR. In this research paper, only two GPCR receptors (MOP and C5a1) were selected for relevant studies. The authors should explain why these two receptors were selected for the study and what their significance is.

3.GPCR is one of the seven - fold transmembrane receptors. As mentioned in this paper, "A Bead-based GPCR phosphorylation immunoassay" can also be called 7TM phosphorylation assay. Therefore, it is suggested that the authors should indicate whether they have studied the other seven transmembrane receptor phosphorylation in addition to GPCR. Moreover, does this experiment also apply to other membrane proteins or even all phosphorylated proteins? I think it is necessary to complete a series of controlled experiments before giving a more accurate and scientific name to this new experimental technique.

4.GPCR phosphorylation is a complex process, and the functions of phosphorylation sites are usually coordinated by multiple sites. However, the technique described in this paper seems to be able to analyze only a single locus. The new technique would be even more attractive if it could detect multiple sites simultaneously.

5.The screening of ligand and GRK inhibitors should be the application prospect of "A Bead-based GPCR phosphorylation immunoassay". However, it seems difficult to cover all three in detail in a single article. In this study, the author can verify and compare the new experimental techniques of this manuscript in more detail and more accurately. It is suggested that the application based on the new technology can be studied separately.

6.It is suggested to objectively evaluate the advantages and disadvantages of "A Bead-based GPCR phosphorylation immunoassay", so as to help readers better understand the possible application scope of this new method.

7.It is suggested that the author compare and analyze "A Bead-based GPCR phosphorylation immunoassay" with existing mature experiments, such as Western Blot, immunoprecipitation, ELISA and other immunological experiments, as well as phosphorylation and modification omics, which will condense the advantages of the new technology.

Response to Reviewers

Black - Reviewer comments

Green – Author comments

Blue – new/edited text

Red – deleted text

Reviewer #1 (Remarks to the Author):

This manuscript by Kaufmann et al. describes the development and validation of a 96 well plate-based immunoassay for detecting and quantifying the phosphorylation of G protein-coupled receptors. It is primarily heterologous in nature, utilizing immunoprecipitation of transfected, epitope-tagged receptors and probing these immunoprecipitates with antibodies raised against phosphopeptides encompassing known phosphorylation sites on the receptor. The manuscript details carefully controlled experiments and thorough validation of the receptor harvesting and detection protocols. The authors successfully applied this approach to the μ -opioid receptor and then to the C5a1 receptor and were able to detect multi-site phosphorylation in both receptors in response to various agonists – moreover, they were able to define the GRKs involved and observed some correlation to the association with β -arrestins. Finally, the authors detailed the utility of this approach to successfully screen for small molecule GRK inhibitors. Overall, this approach is a useful addition to the GPCR signalling/screening armoury and is certainly a useful adjunct/advance over traditional 32P-labelling.

My comments are as follows:

- 1) One obvious limitation of the approach is that the technology is completely dependent on the availability and specificity of phosphorylation-state specific antibodies. This limitation should be explicitly stated in the discussion and some balanced comments added regarding this important challenge as well as some other points raised below.
- 2) The development of such phosphorylation-state specific antibodies cannot be assumed a priori, and many (most) phosphorylation sites have not been empirically determined. Some have been presumed indirectly by mutagenesis and even less is known about the hierarchy and co-operativity of multi-site phosphorylation events. Some recognition of this fact as a major, likely barrier to the broad uptake of this technique should be acknowledged.

Author's reply: In response to Reviewer #1 comment 1 and 2, we have reworded the discussion on page 14, line 293 from bottom: "The assay is versatile, particularly because it can be established for virtually all 7TMRs provided phosphosite-specific antibodies become available. Given the increased number of available phosphosite-specific GPCR antibodies in recent years^{24, 25, 26, 27, 28, 29, 30, 31, 32}, one can envision that 7TM phosphorylation immunoassays can be established for the majority of pharmacologically relevant GPCRs in the near future." We agree with the Reviewer that the technology is completely dependent on the availability of phosphorylation-state specific antibodies. We also agree that this is a major challenge that cannot be accomplished by a small academic laboratory for their individual GPCR of interest. In fact, it is 7TM Antibodies' mission to develop phosphorylation state-specific antibodies for all GPCRs. Over the past two years, such antibodies have been successfully developed for more than 60 GPCRs, and many more are in the pipeline for immediate release. Initial

characterization of all of these antibodies is performed using western blot analysis, which is shown at www.7tmantibodies.com. During assay development, we found that the majority of these antibodies is also well suited for application in 7TM phosphorylation assays. Assays for five GPCRs with chemically diverse ligands are shown in this manuscript. Therefore, we expect that 7TM phosphorylation immunoassays will be established very soon for the majority of pharmacologically relevant GPCRs. In addition to our own efforts, a number of phosphosite-specific antibodies for 7TMRs are already available from other academic and commercial sources, many of them with unknown level of characterization. The assay format presented in this manuscript will be a versatile and less time-consuming procedure to test and validate these antibodies in the future. Moreover, tagged receptor constructs are also available from various commercial sources, so that there will be ample resources and a broad range of applications for future research.

3) In many GPCRs, there are clusters of closely adjacent serines/threonines and it is not clear from the work presented as to whether there is a likelihood that separate antibodies targeting closely position phospho-sites would sterically hinder and compete with each other in this assay if it were to be multiplexed.

Author's reply: In response to Reviewer #1 comment 3, we have deleted "bound to different phosphosite-specific antibodies" from the discussion. Page 15, line 308 from top reads now: "By using fluorescently-labelled beads, it should be possible to establish multiplex applications for quantitative assessment of GPCR phosphorylation patterns in the same sample."

Multiplexing is subject to further assay development and will have to be empirically determined for each individual pair of antibodies. We agree with the Reviewer that the possibility of sterical hinderance between some antibodies targeting closely neighboring phospho-sites has to be considered during development of multiplex applications.

Consequently, we have deleted "bound to different phosphosite-specific antibodies".

Nevertheless, we expect that quantitative assessment of GPCR phosphorylation of the same sample in the same well will at least be possible using combinations of phosphosite-specific and phosphorylation-independent antibodies.

4) The level of heterologous expression was high (300,000 receptors per cell; ~2000 fmol/mg protein) and so some comment on the applicability/sensitivity or otherwise of the technique to eventually be applied to endogenous receptor phosphorylation is warranted (availability of antibody against native receptor notwithstanding).

Author's reply: Response to Reviewer #1 comment 4 and Reviewer #2 comment 3: We agree that the assay was established using a cell line with high expression levels. Receptor expression was quantified using saturation binding experiments in order to calculate the absolute detection limit of the assay. In fact, the assay was more sensitive than western blot and can be performed using stably or transiently transfected cells with lower expression levels. Moreover, an approach to detect endogenous receptor phosphorylation using the novel technology is shown in Supplementary Figure 7. To explicitly state that in the main text, we added the following sentence to the discussion on page 14, line 301 from top: "In fact, the assay is also sensitive enough to detect agonist-induced phosphorylation of endogenously expressed MOP receptors in mouse brain *in vivo*."

Reviewer #2 (Remarks to the Author):

Kaufman et al. describe a new cell-based immunoassay approach to evaluate the ability of the various GRKs to install distinct phosphorylation sites in 7TM receptors, and, based on the robustness of the assay, to screen for novel GRK inhibitors based on the ability of ligands to interfere with phosphorylation at these sites.

The key piece of innovation here is their ability to exploit phosphosite-specific antibodies acting on various GPCRs. They then could use a plate-based immunoassay to detect which sites are labeled under various experimental conditions, and also successfully used the assay to identify which GRKs phosphorylate each specific site in the MOR and other receptors in a quantitative way that cannot be achieved, for example, by phosphoproteomics. The paper is generally well written and figures and tables are for the most part very well put together and easy to interpret.

Enthusiasm is however dampened by the facts that 1) there are far easier and straightforward ways to screen for selective GRK inhibitors, and these have been described now for quite a few years 2) the insular writeup that does not compare and contrast this work with the efforts and products of other labs in developing GRK inhibitors, and 3) the fact that at the end of the day, it is hard to know how much of what was learned about GRK vs. site applies to real physiological processes because these immunoassays are performed in engineered HEK cells overexpressing receptors. We don't know what happens or what GRK has priority in most cell types where the protein concentrations are much lower and not all kinases are equally expressed or available locally. One may indeed use this assay to discover what kinase phosphorylates a GPCR under these experimental conditions, but only at the first site (usually many more sites are modified in a receptor to drive arrestin recruitment). And so this kind of data can only give one a lead for what to look for in vivo.

Author's reply: In response to Reviewer #2 comment 1, we have added the following sentence including relevant references to the results section on page 12, line 248 from bottom: "Previous studies have largely relied on measuring receptor internalization or desensitization in identifying cell-permeable GRK inhibitors^{24, 25, 26, 27, 28, 29}. However, such assays can only indirectly assess GPCR phosphorylation and are difficult to adapt to quantitative high-throughput screening.". Furthermore, we reworded the discussion on page 15, line 328 from top: "However, to date only few potent and selective GRK inhibitors have been discovered^{24, 25, 26, 27, 28, 29, 45}. Among these compound101 is most frequently used to assess the involvement of GRK2/3." and page 16 lines 332 to 337: "We then extended this approach to develop cell-based screening assays for membrane-permeable GRK inhibitors that led to the rapid identification of LDC8988, a novel selective GRK5/6 inhibitor, and LDC9728, a highly potent pan-GRK inhibitor. Both compounds were at least one order of magnitude more potent than compound101. Such novel inhibitors are much-needed tools for elucidating the physiological functions of GRKs." We agree with the Reviewer that efforts and products of other labs in developing GRK inhibitors need to be mentioned and cited.

Author's reply: Response to Reviewer #2 comment 3 and Reviewer #1 comment 4: We agree that the assay was established using a cell line with high expression levels. Receptor expression was quantified using saturation binding experiments in order to calculate the

absolute detection limit of the assay. In fact, the assay was more sensitive than western blot and can be performed using stably or transiently transfected cells with lower expression levels. Moreover, an approach to detect endogenous receptor phosphorylation using the novel technology is shown in Supplementary Figure 7. To explicitly state that in the main text, we added the following sentence to the discussion on page 14, line 301 from top: “In fact, the assay is also sensitive enough to detect agonist-induced phosphorylation of endogenously expressed MOP receptors in mouse brain *in vivo*.”

Reviewer #3 (Remarks to the Author):

The manuscript by Kaufmann et al. describes a method for quantitative evaluation of GPCR phosphorylation based on immunoprecipitation and magnetic bead separation. Firstly, they used magnetic beads to immunoprecipitate the affinity-tagged receptors and then detected the protein using phosphorylation-state-specific and phosphorylation-independent anti-GPCR antibodies. In addition, they demonstrated this concept by treating the two GPCRs with different agonists or antagonists, and generating concentration-response curves. Finally, they established an assay for selective cellular GPCR kinase (GRK) inhibitors, which enabled rapid identification of selective GRK5/6 inhibitors (LDC8988) and high-efficiency PAN GRK inhibitors (LDC9728). This technique can be used as one of the methods to study phosphorylation by GPCR. This study is very interesting and provides new insights into GPCR phosphorylation analysis, which can be widely used for ligand analysis and inhibitor screening. However, there are still some puzzles for the author to solve.

1. The author mentioned several times in this manuscript that a Bead-based GPCR phosphorylation immunoassay can quantitatively analyze the phosphorylation of GPCR, but there seems to be no sufficient evidence to explain this in the manuscript. In the case of quantitative analysis, it should be specified to what extent the technique can pinpoint the magnitude of phosphorylation modification.

Author's reply: The procedure for quantification is described in the methods section on page 26 line 500: “Each sample was divided and placed into two corresponding wells of the assay plate for parallel detection of phosphorylated (80 μ l of lysate per well) and total (40 μ l of lysate per well) receptor levels according to the scheme depicted in Supplementary Fig. 6.” And further on page 27 line 528: “The OD_{405} was then determined using a FlexStation 3 microplate reader (Molecular Devices). Data were acquired with SoftMax Pro 5.4 software and analyzed with Excel 16.0 software. First, the mean of all background controls (without primary antibodies) was subtracted from all values. To normalize the signal intensity, the phosphorylation signal was multiplied by the quotient of the mean of all loading controls divided by the respective loading control. Therefore, the result was adjusted to the amount of receptor for each corresponding sample. This method provides the required information to create quantitative concentration–response curves based on raw OD_{405} values. In each plate, samples of the control agonist were included to facilitate calculation of the results as a percent of agonist control (Supplementary Fig. 1). Values were displayed as concentration-response-curves of at least five independent experiments performed in duplicates and generated with GraphPad Prism 9.3.1 software.” Given the fact that the way of quantification was accepted by Reviewer #1 and Reviewer #2, we feel that it is sufficiently described herein.

2. As we all know, there are more than 800 kinds of GPCR. In this research paper, only two GPCR receptors (MOP and C5a1) were selected for relevant studies. The authors should explain why these two receptors were selected for the study and what their significance is.

Author's reply: In response to Reviewer #3 comment 2, we have now expanded the number of data sets to 5 GPCRs. These data sets were selected to show different GPCRs with chemically diverse ligands. At the same time we took every possible effort to limit the number of different data sets as suggested in comment 5. In addition, we have reworded the discussion on page 14, line 293 from bottom: "The assay is versatile, particularly because it can be established for virtually all 7TMRs provided phosphosite-specific antibodies become available. Given the increased number of available phosphosite-specific GPCR antibodies in recent years^{24, 25, 26, 27, 28, 29, 30, 31, 32}, one can envision that 7TM phosphorylation immunoassays can be established for the majority of pharmacologically relevant GPCRs in the near future." We agree with the Reviewer that the technology is completely dependent on the availability of phosphorylation-state specific antibodies. We also agree that this is a major challenge that cannot be accomplished by a small academic laboratory for their individual GPCR of interest. In fact, it is 7TM Antibodies' mission to develop phosphorylation state-specific antibodies for all GPCRs. Over the past two years, such antibodies have been successfully developed for more than 60 GPCRs, and many more are in the pipeline for immediate release. Initial characterization of all of these antibodies is performed using western blot analysis, which is shown at www.7tmantibodies.com. During assay development, we found that the majority of these antibodies is also well suited for application in 7TM phosphorylation assays. Assays for five GPCRs with chemically diverse ligands are shown in this manuscript. Therefore, we expect that 7TM phosphorylation immunoassays will be established very soon for the majority of pharmacologically relevant GPCRs. In addition to our own efforts, a number of phosphosite-specific antibodies for 7TMRs are already available from other academic and commercial sources, many of them with unknown level of characterization. The assay format presented in this manuscript will be a versatile and less time-consuming procedure to test and validate these antibodies in the future. Moreover, tagged receptor constructs are also available from various commercial sources, so that there will be ample resources and a broad range of applications for future research.

3. GPCR is one of the seven - fold transmembrane receptors. As mentioned in this paper, "A Bead-based GPCR phosphorylation immunoassay" can also be called 7TM phosphorylation assay. Therefore, it is suggested that the authors should indicate whether they have studied the other seven transmembrane receptor phosphorylation in addition to GPCR. Moreover, does this experiment also apply to other membrane proteins or even all phosphorylated proteins? I think it is necessary to complete a series of controlled experiments before giving a more accurate and scientific name to this new experimental technique.

Author's reply: We agree with Reviewer #3 in that we can conclude from our previous and ongoing work on many different receptors that our assay can be adapted to all virtually all 7TMRs and possibly other plasma membrane receptors. Consequently, we have reworded the introduction on page 4 line 74: "Because this assay can be adapted to virtually all seven transmembrane receptors, it is referred to as a '7TM phosphorylation assay'." In fact, as the inventors of a novel assay format, we would like to take the liberty to propose a tentative name.

4. GPCR phosphorylation is a complex process, and the functions of phosphorylation sites are usually coordinated by multiple sites. However, the technique described in this paper seems to be able to analyze only a single locus. The new technique would be even more attractive if it could detect multiple sites simultaneously.

Author's reply: In response to Reviewer #3 comment 3, we have deleted "bound to different phosphosite-specific antibodies" from the discussion. Page 14, line 12 from top reads now: "By using fluorescently-labelled beads, it should be possible to establish multiplex applications for quantitative assessment of GPCR phosphorylation patterns in the same sample." We agree with the Reviewer that there is a high probability of sterical hinderance between antibodies targeting closely neighboring phospho-sites during multiplexing. Consequently, we have deleted "bound to different phosphosite-specific antibodies". Nevertheless, multiplexing is subject to further assay development and will have to be empirically determined for each individual pair of antibodies. We expect that quantitative assessment of GPCR phosphorylation of the same sample in the same well will at least be possible using combinations of phosphosite-specific and phosphorylation-independent antibodies.

5. The screening of ligand and GRK inhibitors should be the application prospect of "A Bead-based GPCR phosphorylation immunoassay". However, it seems difficult to cover all three in detail in a single article. In this study, the author can verify and compare the new experimental techniques of this manuscript in more detail and more accurately. It is suggested that the application based on the new technology can be studied separately.

Author's reply: The focus of this manuscript is the development and application of a novel phosphorylation assay. We limit data presentation to show only three examples of important assay application. It is intended to publish more details in relation to different receptors. In particular, we agree that the structure and full characterization of novel GRK Inhibitors should be published elsewhere.

6. It is suggested to objectively evaluate the advantages and disadvantages of "A Bead-based GPCR phosphorylation immunoassay", so as to help readers better understand the possible application scope of this new method.

7. It is suggested that the author compare and analyze "A Bead-based GPCR phosphorylation immunoassay" with existing mature experiments, such as Western Blot, immunoprecipitation, ELISA and other immunological experiments, as well as phosphorylation and modification omics, which will condense the advantages of the new technology.

Author's reply: A detailed comparison of assay data with western blot results is shown in Figures 2d-f and j-l as well as Supplementary Figures 1 and 2. Advantages and disadvantages of various receptor assays have been elaborated in the introduction, page 3, line 56 – 66, as well as in the discussion, page 14, line 299 to 324. We believe that the advantages of the new technology are clearly presented in the manuscript.

Reviewers' comments:

Reviewer #1 (Remarks to the Author):

My comments have been addressed.

Reviewer #2 (Remarks to the Author):

The authors have addressed most of my concerns in their revision and added additional data. Overall I am supportive of this work, and think it will be a valuable addition to the field. But there are several points that still should be addressed, some minor, one major. All I think are easily corrected.

1) This paper principally concerns two "canonical" GPCRS: MOP and c5a1 and the Abs raised against their GRK-installed phosphosites. Issues are...

a) They seem to definitively say what the potential sites in these receptors are, but there are no references to such determination. The MOP is well studied, but c5a1 is less so. Please cite the key papers/data that show that these are the main/only sites. Otherwise, when the authors state things like their GRK KO cell lines do not phosphorylate a receptor, we only know that they do not phosphorylate the sites their Abs were raised against and they should make that caveat clear.

b) In Fig. s9a, the S/T residues that are phosphorylated (according to the authors) are red, the ones that are not (presumably) are in grey. A description of what grey means exactly should be in the caption. How do we know whether the grey sites are also getting hit or not?

c) The authors call c5a1 "canonical", and I know by this they mean it couples to G proteins and GRKs/arrestins, but when one looks at the structures that have been determined for this GPCR it has a noncanonical arrangement of H8, which seems to auto inhibit the receptor by entering the cytoplasmic cleft. In this configuration, the phosphosites closest to TM7 are not accessible, which makes things less than clear cut when it comes to what is going on at this signature in cells. Do the authors have any thoughts on this? It would help to know how they confirmed what the phosphosites were to understand their line of reasoning.

2) The authors promote their cell-based method as a new way to screen for cell-permeable GRK selective inhibitors. On page 12 they write "Previous studies have largely relied on measuring receptor internalization or desensitization in identifying cell-permeable GRK inhibitors 24, 25, 26, 27, 28, 29. However, such assays can only indirectly assess GPCR phosphorylation and are difficult to adapt to quantitative high-throughput screening". This was a little careless, blanket statement about these references within the context of this paragraph. While some of these referenced papers do have experiments looking at internalization/desensitization of their hits, that was not typically the basis of their screens.

Of course the big advantage of the new screen is that one can more efficiently screen in a living cell (although important to note it is not a physiologically relevant cell line) and evaluate selectivity by directly reading out what happens at some phosphosites. But like any cell based screen, it still suffers the ambiguity of knowing whether the compounds are hitting the target. The way the screen is set up to look at GRK dependent phosphorylation reduces ambiguity, but doesn't eliminate other possibilities (just throwing stuff out here) such as disrupting localization of GPCRs or GRKs, or other regulatory factors specific for the phosphosites in question.

What I am getting at is that the authors could be a little more holistic about the pros and cons of the new screen. If you screen against purified proteins, you know quantitatively what your selectivity and potency is. In cells, you really cannot ever be sure without testing the purified proteins (as they do

with their hits here). With purified proteins, you don't know if you will have off-target effects in cells, so one goes and tests that later. In the proposed assay, you still don't, because one is only evaluating whether phosphorylation is affected. The fact that one needs specific Abs for confirmed phosphosites is probably the biggest hurdle for others planning to do such screens, and perhaps this is not a huge issue given the company behind this effort, but in going back to the first point above about knowing what the relevant phosphosites are in the first place, in every signaling context, in every cell type, is a huge hurdle. So it has its advantages, but there are limitations too. Just like any other screen. It might be the best screen to use in some scenarios, but not others.

3) My biggest problem, however is that, as stated in the abstract, this assay has been used for "rapid identification of a selective GRK5/6 inhibitor (LDC8988) and a highly potent pan-GRK inhibitor (LDC9728)."

There are literally only 8 lines of text devoted to this important, abstract-worthy result. There are no methods for it. We don't know if it was "rapid" or not. We don't know statistics behind the screen like Z prime scores. We know nothing about the chemical nature of these compounds or really how good of hits they were in the screen or even how unique they are from other scaffolds.

I know the authors plan to publish the screen elsewhere, but we have no manuscript here to look at and have no methods to evaluate and so I do not think it is appropriate to include data with LDC8988 or LDC9728. Because they likely would have the same analysis for these hits in their screening paper, it also potentially means publishing the same results twice. So I recommend that the authors should stop their data here with compound101 as proof of principle because that compound is reasonably well documented. It would have been nice to try some of the other lead hits, like paroxetine or 14as in the Waldschmidt paper that are clearly membrane permeable (compound101 is not considered bioavailable), but this not required for my approval.

Probably the first time I have ever asked authors to remove data from a paper!

4) Minor editorial points:

a) "Western" should not be capitalized.

b) The significant figures need to be consistent and realistic in the data tables and text throughout the manuscript. For example, in Table S3, the number of significant figures for the GRK2 assay range from 2 to 4 for the IC50 value, and 2 to 3 for the error. Based on my experience, the authors are very unlikely to obtain "1.519 +/- 0.213" again if they were to repeat this experiment. It is more likely a 2 significant figure data point (i.e. "1.5 +/- 0.2"). By being realistic about the accuracy of the numbers, it also makes it easier for readers to evaluate differences in the table entries because only the relevant digits are shown.

Another example is in Table S2 where it is formatted to look pretty (with the number of decimal places constant, etc.) but it is not realistic. The first pEC50 value is probably 6.3 +/- 0.1. That is much more repeatable than 6.31 is considering the error is 0.1. Plus then the reader will see values like 6.3, 6.3 and 6.2 for that first column of data and it is easy to see that they are not statistically different considering the error. Another problem in this table is that the Emax is reported as "100" with no error (1 significant figure) whereas the Emax in the last column has 4 significant figures (also no errors). I know the 100 comes from normalization, but standard practice is report error terms for these Emax values based on duplicates in the assays. Please consider correcting such problems throughout the manuscript.

John Tesmer

Reviewer #3 (Remarks to the Author):

After reading the response letter and revised manuscript, the author basically answered all my questions. Compared with the previous draft, this paper has been significantly improved. But two small questions remain unanswered.

1. It is hoped that there will be direct evidence to prove that this new technique can accurately quantify the level of receptor phosphorylation modification (pM or fM ?), rather than explain how to quantify phosphorylation of receptors.

2. It is suggested that the author explain why MOP and C5a1 receptors were selected for the study, rather than simply increasing the number of GPCR used in the experiment from two to five. There is a large number of GPCR, and without screening criteria, it is meaningless to simply increase the number of receptors used in experiments.

Response to Reviewers

Black - Reviewer comments

Red – Author comments

Blue – new/edited text

Reviewer #1 (Remarks to the Author):

My comments have been addressed.

Reviewer #2 (Remarks to the Author):

The authors have addressed most of my concerns in their revision and added additional data. Overall I am supportive of this work, and think it will be a valuable addition to the field. But there are several points that still should be addressed, some minor, one major. All I think are easily corrected.

1) This paper principally concerns two "canonical" GPCRs: MOP and c5a1 and the Abs raised against their GRK-installed phosphosites. Issues are...

a) They seem to definitively say what the potential sites in these receptors are, but there are no references to such determination. The MOP is well studied, but c5a1 is less so. Please cite the key papers/data that show that these are the main/only sites. Otherwise, when the authors state things like their GRK KO cell lines do not phosphorylate a receptor, we only know that they do not phosphorylate the sites their Abs were raised against and they should make that caveat clear.

In response to the reviewer, we have cited studies on MOP phosphorylation sites and provide more details on identification of C5a1 sites page 9, line 180: "MOP is a primary target for opioid analgesics. MOP desensitization and tolerance are regulated by phosphorylation of four carboxyl-terminal serine and threonine residues, namely, T370, T376 and T379 in addition to S375^{19, 20}." and page 11, line 221: "Next, we tested whether phosphorylation assays can be used for assessing GPCRs, which predominantly undergo GRK5/6-dependent phosphorylation. The complement component 5a receptor 1 (C5a1) is involved in host response to infection and tissue damage. Unlike MOP, C5a1 seems to require primarily GRK5/6 for β -arrestin recruitment^{7, 23}, however, the exact contribution of individual GRK isoforms to C5a1 phosphorylation remains unclear. The generation of phosphosite-specific C5a1 antibodies revealed that at least 8 out of 11 S/T residues within the carboxyl-terminal tail undergo agonist-driven phosphorylation. Antibodies targeting three distinct phosphorylation clusters were then extensively characterized and selected for the development of the corresponding pT324/pS327-, pS332/pS334-, pS338/pT339-C5a1 phosphorylation assays (Supplementary Fig. 9-11)." The fact that the assay detects phosphorylation only when antibodies are available is clearly stated in the manuscript. See page 14 line 304: "The assay is versatile, particularly because it can be established for virtually all 7TMRs provided phosphosite-specific antibodies become available."

b) In Fig. s9a, the S/T residues that are phosphorylated (according to the authors) are red, the ones that are not (presumably) are in grey. A description of what grey means exactly should be in the caption. How do we know whether the grey sites are also getting hit or not?

We have added a description of what grey means to the caption of Figures 5 and S9.

c) The authors call c5a1 "canonical", and I know by this they mean it couples to G proteins and GRKs/arrestins, but when one looks at the structures that have been determined for this GPCR it has a noncanonical arrangement of H8, which seems to auto inhibit the receptor by entering the cytoplasmic cleft. In this configuration, the phosphosites closest to TM7 are not accessible, which makes things less than clear cut when it comes to what is going on at this signature in cells. Do the authors have any thoughts on this? It would help to know how they confirmed what the phosphosites were to understand their line of reasoning.

We assume that the reviewer referred to the C5a1 structure reported in Robertson et al. Nature 553:111-114, 2018. As with many GPCRs, this structure lacks much of the carboxyl terminus, including the majority of potential phosphate acceptor sites. In addition, the receptor is bound to an inverse agonist that does not induce receptor phosphorylation. 7TM Antibodies uses proprietary technology (EP22167875) to identify relevant phosphorylation sites on GPCRs with high success rate. Analysis of currently available GPCR structures is not part of this process.

2) The authors promote their cell-based method as a new way to screen for cell-permeable GRK selective inhibitors. On page 12 they write "Previous studies have largely relied on measuring receptor internalization or desensitization in identifying cell-permeable GRK inhibitors 24, 25, 26, 27, 28, 29. However, such assays can only indirectly assess GPCR phosphorylation and are difficult to adapt to quantitative high-throughput screening". This was a little careless, blanket statement about these references within the context of this paragraph. While some of these referenced papers do have experiments looking at internalization/desensitization of their hits, that was not typically the basis of their screens. Of course the big advantage of the new screen is that one can more efficiently screen in a living cell (although important to note it is not a physiologically relevant cell line) and evaluate selectivity by directly reading out what happens at some phosphosites. But like any cell based screen, it still suffers the ambiguity of knowing whether the compounds are hitting the target. The way the screen is set up to look at GRK dependent phosphorylation reduces ambiguity, but doesn't eliminate other possibilities (just throwing stuff out here) such as disrupting localization of GPCRs or GRKs, or other regulatory factors specific for the phosphosites in question. What I am getting at is that the authors could be a little more holistic about the pros and cons of the new screen. If you screen against purified proteins, you know quantitatively what your selectivity and potency is. In cells, you really cannot ever be sure without testing the purified proteins (as they do with their hits here). With purified proteins, you don't know if you will have off-target effects in cells, so one goes and tests that later. In the proposed assay, you still don't, because one is only evaluating whether phosphorylation is affected. The fact that one needs specific Abs for confirmed phosphosites is probably the biggest hurdle for others planning to do such screens, and perhaps this is not a huge issue given the company behind this effort, but in going back to the first point above about knowing what the relevant phosphosites are in the first place, in every signaling context, in

every cell type, is a huge hurdle. So it has its advantages, but there are limitations too. Just like any other screen. It might be the best screen to use in some scenarios, but not others.

In response to the reviewer, we added more information about the process of GRK inhibitor identification in order to clarify that a combination of in vitro GRK assays and cell-based assays was used indeed (see below).

We then attempted to identify selective GRK5 inhibitors that display novel chemotypes. To this end, a library of 16,666 structurally distinct compounds was tested in vitro in GRK5 kinase assays, resulting in a total of 208 primary hits. Based on these hits more than one hundred novel analogs were synthesized in several rounds of optimization. The most potent compounds were then further tested in our cell-based GRK inhibitor assays,..."

3) My biggest problem, however is that, as stated in the abstract, this assay has been used for "rapid identification of a selective GRK5/6 inhibitor (LDC8988) and a highly potent pan-GRK inhibitor (LDC9728)." There are literally only 8 lines of text devoted to this important, abstract-worthy result. There are no methods for it. We don't know if it was "rapid" or not. We don't know statistics behind the screen like Z prime scores. We know nothing about the chemical nature of these compounds or really how good of hits they were in the screen or even how unique they are from other scaffolds. I know the authors plan to publish the screen elsewhere, but we have no manuscript here to look at and have no methods to evaluate and so I do not think it is appropriate to include data with LDC8988 or LDC9728. Because they likely would have the same analysis for these hits in their screening paper, it also potentially means publishing the same results twice. So I recommend that the authors should stop their data here with compound101 as proof of principle because that compound is reasonably well documented. It would have been nice to try some of the other lead hits, like paroxetine or 14a in the Waldschmidt paper that are clearly membrane permeable (compound101 is not considered bioavailable), but this not required for my approval. Probably the first time I have ever asked authors to remove data from a paper!

In response to the reviewer, we added more information about the process of GRK inhibitor identification on page 13 lines 270: "We then attempted to identify selective GRK5 inhibitors that display novel chemotypes. To this end, a sublibrary of 16,666 structurally distinct potential kinase inhibitors was tested in vitro in GRK5 kinase assays, resulting in a total of 208 primary hits. We independently confirmed and validated the most potent primary GRK5 hits, followed by hit exploration and medicinal chemistry-based optimization of a genuinely known compound class of ATP competitive kinase inhibitors although not previously described as GRK inhibitors. Hit exploration and the generation of novel analogs quickly led to the identification of more potent and largely double digit nanomolar GRK5 inhibitors as well as to an understanding of the structure-activity-relationship. Ten of such novel GRK5 inhibitor analogs were selected for further testing in our cell-based GRK inhibitor assays, which led ..." These novel inhibitors are tool compounds which will facilitate elucidation of physiological and pathophysiological roles of GRKs. There is clearly an urgent need for such novel research tools. Development of clinical candidates is ongoing. Thus, structures cannot be disclosed here. While we appreciate the reviewers' thoughtful comments and recommendations, we strongly feel that the actual identification of novel GRK inhibitors will provide a much more compelling showcase to demonstrate the validity of this novel assay. We thank the reviewer for his help to strengthen the manuscript as well as for his statement that the removal of

data will NOT be required for his final approval thus helping to facilitate scientific progress in the field of GRK inhibitors.

4) Minor editorial points:

a) "Western" should not be capitalized.

We have corrected this throughout manuscript.

b) The significant figures need to be consistent and realistic in the data tables and text throughout the manuscript. For example, in Table S3, the number of significant figures for the GRK2 assay range from 2 to 4 for the IC50 value, and 2 to 3 for the error. Based on my experience, the authors are very unlikely to obtain "1.519 +/- 0.213" again if they were to repeat this experiment. It is more likely a 2 significant figure data point (i.e. "1.5 +/- 0.2"). By being realistic about the accuracy of the numbers, it also makes it easier for readers to evaluate differences in the table entries because only the relevant digits are shown. Another example is in Table S2 where it is formatted to look pretty (with the number of decimal places constant, etc.) but it is not realistic. The first pEC50 value is probably 6.3 +/- 0.1. That is much more repeatable than 6.31 is considering the error is 0.1. Plus then the reader will see values like 6.3, 6.3 and 6.2 for that first column of data and it is easy to see that they are not statistically different considering the error. Another problem in this table is that the Emax is reported as "100" with no error (1 significant figure) whereas the Emax in the last column has 4 significant figures (also no errors). I know the 100 comes from normalization, but standard practice is report error terms for these Emax values based on duplicates in the assays. Please consider correcting such problems throughout the manuscript.

As requested by the reviewer pEC50 values were displayed as 2 significant figure data points and Emax values were reported with error terms throughout manuscript.

Reviewer #3 (Remarks to the Author):

After reading the response letter and revised manuscript, the author basically answered all my questions. Compared with the previous draft, this paper has been significantly improved. But two small questions remain unanswered.

1. It is hoped that there will be direct evidence to prove that this new technique can accurately quantify the level of receptor phosphorylation modification (pM or fM ?) rather than explain how to quantify phosphorylation of receptors.

Yes, the new technique can accurately quantify the level of receptor phosphorylation modification if number of receptors per cell are known from binding studies. This is described in the results on page 8 line 159 and depicted in Figure S4: "Saturation-binding studies revealed that the MOP-HEK293 cells expressed 300,000 functional MOPs per cell (Supplementary Fig. 3), allowing us to calculate a detection limit of 80 pg receptor protein for the pS375-MOP immunoassay and 500 pg for the immunoblot assay. In the range of 80 pg to 1200 pg receptor protein, the pS375-MOP immunoassay showed a linear increase in signal intensity (Supplementary Fig. 4)."

2. It is suggested that the author explain why MOP and C5α1 receptors were selected for the study, rather than simply increasing the number of GPCR used in the experiment from two to five. There is a large number of GPCR, and without screening criteria, it is meaningless to simply increase the number of receptors used in experiments.

In response to the reviewer, we have added information why MOP and C5a1 were selected page 9, line 180: "MOP is a primary target for opioid analgesics. MOP desensitization and tolerance are regulated by phosphorylation of four carboxyl-terminal serine and threonine residues, namely, T370, T376 and T379 in addition to S375^{19, 20}." and page 11, line 221: "Next, we tested whether phosphorylation assays can be used for assessing GPCRs, which predominantly undergo GRK5/6-dependent phosphorylation. The complement component 5a receptor 1 (C5a1) is involved in host response to infection and tissue damage. Unlike MOP, C5a1 seems to require primarily GRK5/6 for β-arrestin recruitment^{7, 23}, however, the exact contribution of individual GRK isoforms to C5a1 phosphorylation remains unclear. We also provide a rational basis for selecting additional receptors: "The final experiments were designed to assess the utility of available phosphosite-specific antibodies for the development of phosphorylation assays for other GPCR targets with chemically diverse ligands."

EVIEWERS' COMMENTS:

Reviewer #2 (Remarks to the Author):

My comments have been adequately addressed.

Reviewer #3 (Remarks to the Author):

The authors have answered most of my questions, and the conclusions are now more solid. However, I still have a question for the author to answer.

The author describes in the article, "Saturation-binding studies revealed that the MOP-HeK293 cells expressed 300,000 functional MOPs per cells (Supplementary Fig. 3) ", but in Supplementary Fig. 3, there seems to be no direct information showing that each cell expressed 300,000 MOPs. According to the suggestions in the article, there is still no method to detect the number of receptors expressed in a single cell in the method section. If the author can answer this technical question, the conclusion of the article will be more solid.

Response to Reviewers

Black - Reviewer comments

Green – Author comments

Blue – new/edited text

REVIEWERS' COMMENTS:

Reviewer #2 (Remarks to the Author):

My comments have been adequately addressed.

Reviewer #3 (Remarks to the Author):

The authors have answered most of my questions, and the conclusions are now more solid. However, I still have a question for the author to answer.

The author describes in the article, "Saturation-binding studies revealed that the MOP-HeK293 cells expressed 300,000 functional MOPs per cells (Supplementary Fig. 3)", but in Supplementary Fig. 3, there seems to be no direct information showing that each cell expressed 300,000 MOPs. According to the suggestions in the article, there is still no method to detect the number of receptors expressed in a single cell in the method section. If the author can answer this technical question, the conclusion of the article will be more solid.

Author's reply: In response to Reviewer #3 comments, we have added the following text on page 8 lin 159: Saturation radioligand binding revealed 2279 fmol specific binding sites per mg membrane protein (Supplementary Fig. 3). Based on protein assays 1 mg membrane protein was calculated to represent 4.6×10^6 cells. Thus, 2279 fmol equal $2279 \times 6 \times 10^8$ receptors per mg membrane protein, representing 4.6×10^6 cells, which results in 297,260 (~300,000) functional MOP receptors per cell.